



# Regional influence of ocean-climate teleconnections on the timing and duration of MODIS derived snow cover in British Columbia, Canada

Alexandre R. Bevington[1,2], Hunter E. Gleason[1], Vanessa N. Foord[1], William C. Floyd[3,4], and Hardy P. Griesbauer[1]

[1]British Columbia Ministry of Forests, Lands, Natural Resource Operations and Rural Development, Prince George, V2L 1R5, Canada;
[2]Natural Resources and Environmental Studies Institute and Geography Program, University of Northern British Columbia, Prince George, V2N 4Z9, Canada;
[3]British Columbia Ministry of Forests, Lands, Natural Resource Operations and Rural Development, Nanaimo, V9T 6E9, Canada;
[4]Vancouver Island University, Nanaimo, V9R 5S5, Canada;

**Correspondence:** Alexandre R. Bevington (alexandre.bevington@gov.bc.ca)

**Abstract.** We use the twice daily Moderate Resolution Imaging Spectro–Radiometer (MODIS) snow cover product to study the regional influence of the Oceanic Niño Index (ONI) and Pacific Decadal Oscillation (PDO) on snow cover in British Columbia (BC). We apply a locally weighted regression (LOWESS) interpolation to the MODIS normalized difference snow index (NDSI) time series to detect the timing and duration of snow. We confirm the general consensus from many previous *in situ* studies that both ONI and PDO have significant impacts on snow cover in BC. We add to this knowledge by performing seasonal and regional analysis using established hydrozones, and explore variation in our results by elevation bins of 500 m. We calibrated our method with *in situ* snow water equivalent (SWE) data, and found an optimal NDSI threshold of 30 for detecting continuous snow cover. We separate automatic snow weather station data into calibration (75%) and validation (25%) subsets and obtain mean absolute errors between the MODIS and *in situ* observations for the start, end and duration of 8.7, 8.9 and 13.1 days for the calibration data, and 12.7, 12.6 and 16.6 for the validation data, respectively. In general, the start date of snow is poorly correlated with both ONI and PDO, whereas end date and duration are strongly negatively correlated. Regional patterns emerge where northern and southern BC are most correlated to the PDO and the ONI, respectively. These relationships are generally stronger at lower elevations, and vary spatially. This study demonstrates that the suitability of ocean-climate teleconnections as predictors of the timing and duration of snow varies throughout BC.

## 1 Introduction

The timing and duration of seasonal snow cover have been shown to influence a number of landscape processes, namely soil moisture and drought (e.g., Fang and Pomeroy, 2008), wildfire activity and intensity (e.g., Westerling et al., 2006), freshwater availability (e.g., Barnett et al., 2005), flooding (e.g., Barnett et al., 2005), plant ecology (e.g., Jonas et al., 2008), and wildlife behavior (e.g., Boelman et al., 2019). The Western Canadian province of British Columbia (BC) is roughly one mil-



lion square kilometers in size, has multiple mountain chains, and shares its border with 1 province, 2 territories and 4 states (Fig. 1). Generally mild and dry winters in BC have been identified as important factors in forest insect outbreaks (Stahl et al., 2006b), wildfire activity (Crowley et al., 2019), glacier change (e.g., Menounos et al., 2018) and overall drought conditions (Trishchenko et al., 2016). The winter snowpack differs in timing and duration throughout the province and is influenced by,

among other processes, climate change (e.g., Allchin and Déry, 2017; Foord, 2016; Najafi et al., 2017), local geography (e.g., Moore and McKendry, 1996; Schnorbus et al., 2014; Stahl et al., 2006a), land cover change (e.g., Boon, 2012), and ocean–climate teleconnections (e.g., Fleming and Whitfield, 2010; Hsieh and Tang, 2001; Shabbar, 2006; Winkler and Moore, 2006). Observed snow water equivalent (SWE) in southern BC has significantly decreased by up to -10 % per decade between 1950 and 2014 (e.g., White et al., 2016). Looking further back, Foord (2016) studied the historical instrumented climate record

(∼100 years) in northern BC and found significant increases in winter air temperatures and decreases in winter precipitation as snow. Similarly, 90 % of snow monitoring sites across the Western US have recently shown declines (Mote et al., 2018). These studies rely on *in situ* observations with important limitations, e.g., the under representation of alpine environments, the sparse network of snow observations in western North America, and the challenges of interpreting *in situ* observations across broad heterogeneous mountainous landscapes.

The Moderate Resolution Imaging Spectro–Radiometer (MODIS) optical imaging sensors aboard the Terra (launched in 1999) and Aqua (launched in 2002) satellite platforms provide a nearly continuous record of twice daily global daytime 500 m resolution satellite images. MODIS has been used to study different aspects of snow cover in, for example, the Andes (Saavedra et al., 2017, 2018), Alaska (Lindsay et al., 2015), Asia (Li et al., 2018; Tang et al., 2017), British Columbia (Trubilowicz et al., 2016), Canada (Trishchenko et al., 2016) and more recently for global studies (e.g., Hammond et al., 2018; Bormann et al.,

2018). Although the spatial extent and duration of snow cover are well studied, there have been relatively few studies that have utilized MODIS to detect the start ($SD_{ON}$) and end ($SD_{OFF}$) dates of snow cover (e.g., Li et al., 2018; Lindsay et al., 2015). Lindsay et al. (2015) were able to demonstrate using MODIS that nearly half of Alaska and surrounding areas experienced intermittent snow–covered periods with field validated accuracies between -12.2 and +33.9 days. Saavedra et al. (2018) used similar methods in the Andes Mountains and found negligible trends in snow persistence (SP) over time, however SP was

significantly correlated with temperature (negative correlation) and precipitation (positive correlation). In British Columbia, basin scale investigations of MODIS derived snow characteristics were used as a predictor of river streamflow (Tong et al., 2009). They found that the 50 % snow cover fraction value was highly correlated with the timing of 50 % normalized accumulated runoff ($r_S = 0.82$; p < 0.001) in the Quesnel River Basin, a sub-boreal mountainous watershed. Finally, Trishchenko et al. (2016) computed the minimum snow and ice cover extents over Canada and found significant discrepancies with global

land cover products. This growing body of research demonstrates the wide variety of applications of the MODIS snow cover product.

The El Niño Southern Oscillation (ENSO) and the Pacific Decadal Oscillation (PDO) have been identified as the two most influential ocean-climate teleconnections on snow cover in BC (Moore and McKendry, 1996). This paper investigates the regional effects of ENSO and PDO on the MODIS derived $SD_{ON}$ and $SD_{OFF}$ and the total duration of snow cover ($SD_{DUR}$) in

BC. To do this, we leverage daily MODIS snow cover data (Riggs and Hall, 2015) using Google Earth Engine (Gorelick et al.,





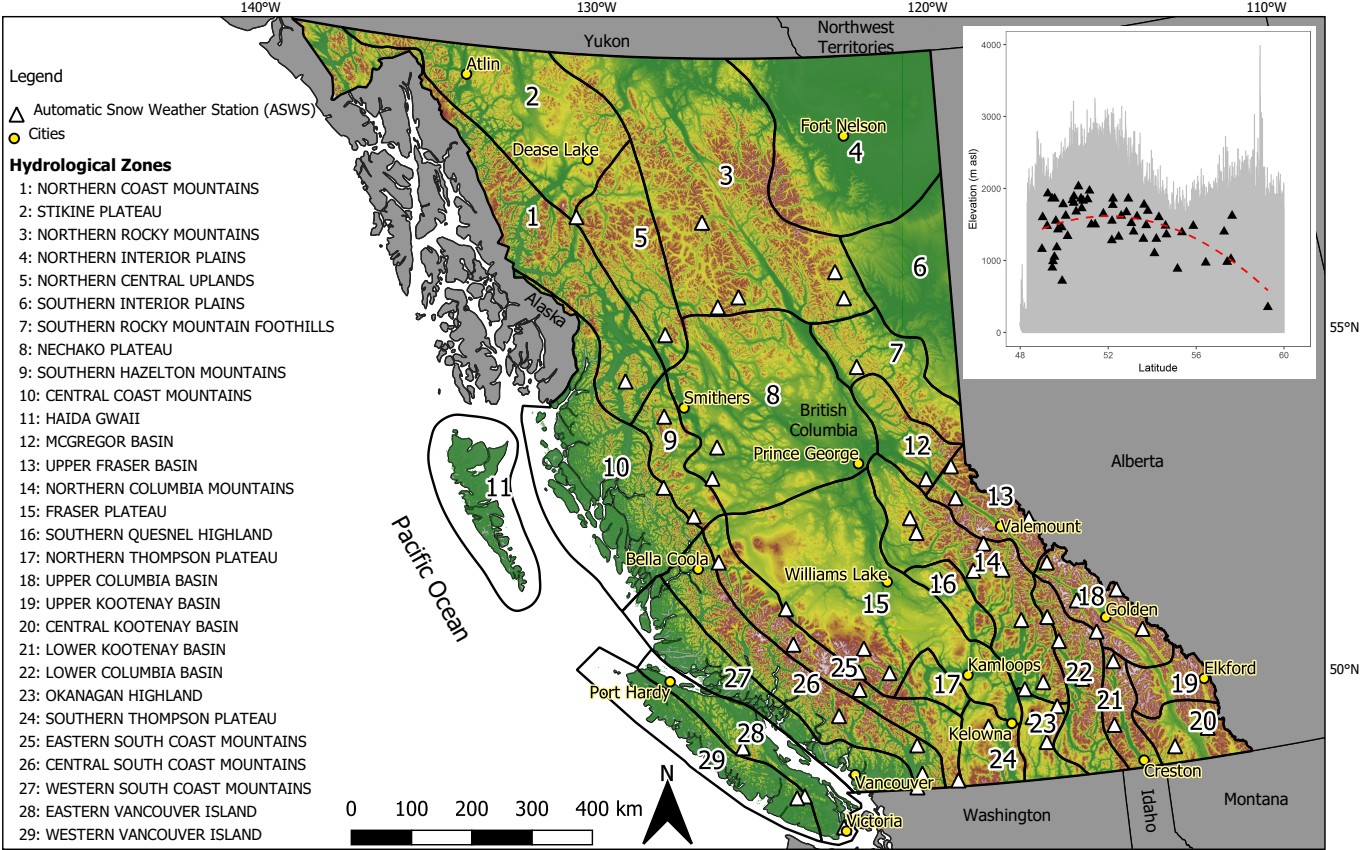

**Figure 1.** Map of British Columbia, Canada, shown with hydrozones, automated snow weather stations (ASWS), and selected cities. Inset shows the distribution of ASWS locations by latitude (x-axis) and elevation (y-axis) plotted against terrain elevation (grey line) with a polynomial fit to the data (dashed red line) indicating gaps in *in situ* observations in northern BC, particularly at high elevation.

2017). We then apply a Locally Weighted Scatterplot Smoothing (LOWESS) interpolation to the time series and detect $SD_{ON}$, $SD_{OFF}$ and $SD_{DUR}$ for each hydrological year (1 Sep–1 Sep) between 2002–2018. We calibrate and validate our workflow to a province–wide snow observation network (Fig. 1). We acknowledge that this is a short time period (16 years), and suggest caution in the implementation of our results beyond the period of study. Furthermore, a thorough explanation of the climatology and physical processes that control ocean–climate teleconnections and their regional influence on snow is outside the scope of this paper.

## 2 Ocean–climate teleconnections and the snowpack of British Columbia

Hemispheric–scale circulation patterns, or ocean–climate teleconnections, are known to influence the climatology of Western Canada, the most influential patterns include: PDO (Stahl et al., 2006a; Whitfield et al., 2010), ENSO (Moore and McKendry,





1996; Shabbar, 2006), Pacific North American pattern (PNA; Bonsal and Shabbar, 2008; Stahl et al., 2006a), and Arctic Oscillation (AO; Burn, 2008; Fleming et al., 2016). Anthropogenic climate change is also an important influence on snow cover in BC (e.g., White et al., 2016; Foord, 2016; Islam et al., 2017; Najafi et al., 2017; Schnorbus et al., 2014), however, climate change studies require long time series and are complex due to the potential influence of climate change on the ocean–
climate teleconnections themselves (e.g., Cai et al., 2018).

Over Western Canada, the PDO and ENSO influence the frequency of major synoptic circulation types associated with storms generated from the North Pacific Ocean (Stahl et al., 2006a). This also affects the Western United States, where Svoma (2011) found that the warm phase of ENSO (El Niño) years had higher snowlines, with the strongest response near the Pacific Coast. Bonsal and Shabbar (2008) provide a thorough review of relationships between large-scale teleconnections and low water flows
over Canada, with a section dedicated to Western Canada.

## 3   Datasets

### 3.1   Ocean–climate teleconnections

The ENSO is measured as anomalous sea surface temperatures (SST) over the eastern equatorial Pacific Ocean with corresponding changes in the atmospheric sea-level pressure in the tropical Pacific (Shabbar, 2006). The diabatic heating over the
tropical Pacific in response to El Niño initiates Rossby waves that enhance PNA circulation over North America, the most pronounced cell of the PNA lies over Western Canada (Wallace and Gutzler, 1981). In this study, we test the correlation of $SD_{ON}$, $SD_{OFF}$ and $SD_{DUR}$ in BC with the Oceanic Niño Index (ONI; Fig. 2), a measurement of ENSO strength (Huang et al., 2017). The ONI index is calculated as the 3-month running mean of Extended Reconstructed Version 5 SST anomalies in the Niño 3.4 (5° N–5° S, 120° E–170° W) region (Huang et al., 2017). ONI events are defined as periods where the index exceeds
0.5 °C (El Niño) or is below -0.5 °C (La Niña) for more than 5 consecutive months (Trenberth, 1997).

A longer lived mode of climate variability known to influence the hydro–climatology of Western Canada is the PDO (Fig. 2). Like ENSO, the PDO is an organized mode of low-frequency ocean–atmosphere circulation, unlike ENSO, the PDO manifests as more of a regime shift, occurring at roughly decadal intervals (Whitfield et al., 2010). The origins and patterns of the PDO are poorly understood in comparison to the mechanisms of ENSO (Whitfield et al., 2010). An index defining the PDO was first
proposed by Mantua et al. (1997), it is calculated as the leading principle component from an un-rotated empirical orthogonal function analysis of monthly SST anomalies poleward of 20° N for the period 1900–1993 (Whitfield et al., 2010). The PDO index has been shown to have strong links with sea-level pressure and surface wind stress (Mantua et al., 1997). Stronger more coherent events may occur when the PDO phase and ENSO phase coincide (Gershunov and Barnett, 1998). Similarly to the ONI, we define PDO events of 5 consecutive months with values greater than 0.5 °C as positive (warm) and below -0.5 °C as
negative (cool) phases (Kiem et al., 2003).

In this study, we summarize the seasonal teleconnection values of the ONI and PDO (Fig. 2). We define the seasons as Spring (March–May), Summer (June–August), Fall (September–November) and Winter (December–February). For example, the teleconnection indices used in this study for a given year start in March of that year, until February of the following year. Over our





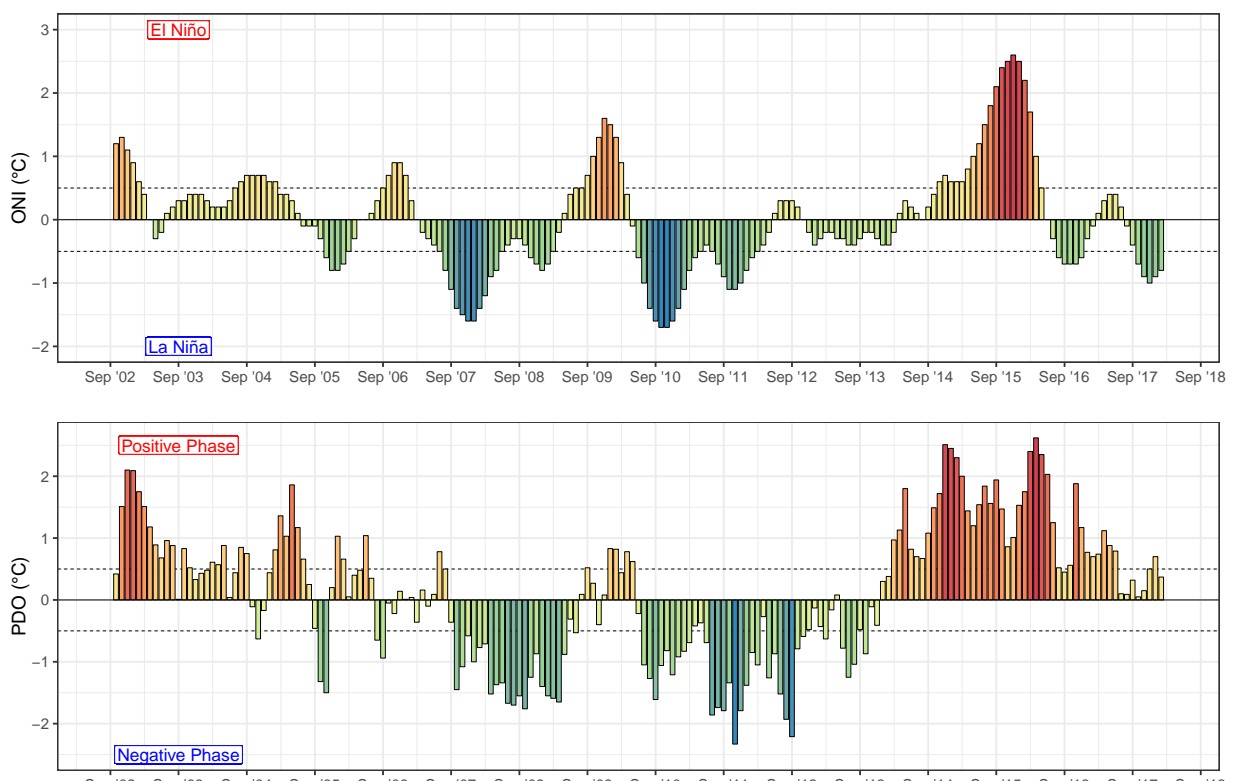

**Figure 2.** Monthly values of Oceanic Niño Index (ONI; top panel; National Oceanic and Atmospheric Adminsitration, 2019), and Pacific Decadal Oscillation (PDO; lower panel; Joint Institute for the Study of the Atmosphere and Ocean, 2019) over time (2002-2018). ONI and PDO phases are generally considered in phase when anomalies are over 0.5 °C or under -0.5 °C (dashed grey lines) for 5 consecutive months, the ±0.5 °C line is marked as a dashed grey line.

short period of interest, the ONI and PDO are well correlated (Spearman Correlation = 0.56, p < 0.001) and the ONI serves as a better predictor of PDO (0.74 °C °C $^{-1}$, $r^2$ = 0.91) than the PDO of ONI (0.42 °C °C $^{-1}$, $r^2$ = 0.69).

### 3.2 MODIS snow cover product

We combine the daily daytime 500 m MODIS C6 Snow Cover Product from MODIS Aqua (MYD10A1) and MODIS Terra
5 (MOD10A1) in BC, termed M*D10A1 (Hall et al., 2002; Riggs and Hall, 2015). We only use imagery acquired after 1 Sep 2002 as this is the first hydrological year where both platforms are fully operational. M*D10A1 measures presence of snow cover using the normalized difference snow index (NDSI; Hall et al., 2011), see Eq.( 1) below:

$$NDSI = \frac{G - SWIR}{G + SWIR} \tag{1}$$



Where *G* (green) represents MODIS band 4 (0.545–0.565 μm) and *SWIR* (shortwave infrared) represents MODIS band 6 (1.628–1652 μm). Fresh snow on land is highly reflective in the visible wavelengths and poorly reflective in the shortwave infrared wavelengths. The magnitude of this difference is exploited by the NDSI and normalized between -1 and 1, where pixels with NDSI > 0 have snow present, and those ≤ 0 are snow free Riggs and Hall (2015). Additionally, a series of data

screens are applied to allow quality control of the NDSI results, these include data flags stored in the product as a quality assurance band (Riggs and Hall, 2015). The resulting NDSI snow cover product is scaled between 0–100 %, where 100 % represents full snow coverage. We removed water and lakes from the study area using the MOD44W V6 250 m Terra Land Water Mask (Carroll et al., 2017). Mean elevation was calculated for every MODIS pixel from the higher 30 m resolution Shuttle Radar Topography Mission (SRTM; Farr et al., 2007). An increasing amount of studies are using M*D10A1 to study

snow cover as the dataset is approaching 20 years. Most, however use an 8 or 16 day aggregated product in order to temporally filter the influence of clouds.

### 3.3 Hydrozones

We use BC Government hydrozone boundaries to divide our study area into 29 units (Fig. 1). This data was accessed through the 'bcmaps' R-language package (Teucher et al., 2018). The hydrozones define areas that are hydrologically and geomor-

phologically unique based on precipitation patterns, low flows, peak flows and underlying surficial geology (Coulson and Obedkoff, 1998). The inherent link of the hydrozones with regional hydrology make them a logical spatial unit by which to investigate regional variation in snow duration and timing in BC.

### 3.4 Observation network

Automatic snow and weather station (ASWS; Fig. 1) daily observations (BCMOE, 2019) were used to calibrate the remotely

sensed NDSI time series to the ASWS SWE time series. These stations are generally located near treeline and are primarily used for river forecasting. The SWE time series was used to determine annual snow duration at each station for years with valid data between 2002–2018. Of the 99 available stations, only 60 had data that corresponded with the MODIS period of record. The 60 stations are located between 350–2030 m above sea level (asl) with an average elevation of 1485 m.

## 4 Methods

We define $SD_{DUR}$ as the difference between $SD_{OFF}$ and $SD_{ON}$. Our workflow consists of 1) MODIS pre-processing; 2) snow season extraction, and 3) data output. Additionally we calibrate and validate our workflow with ASWS data (Fig. 3).

### 4.1 MODIS pre-processing

Google Earth Engine (GEE) was used to combine daily 500 m MODIS snow cover products from MOD10A1 and MYD10A1, termed M*D10A1. For each day, the maximum NDSI between MOD10A1 and MYD10A1 was calculated per pixel. The daily

time series (2002–2018) was then extracted at every ASWS location and at 65 thousand random locations. Cloudy pixels are





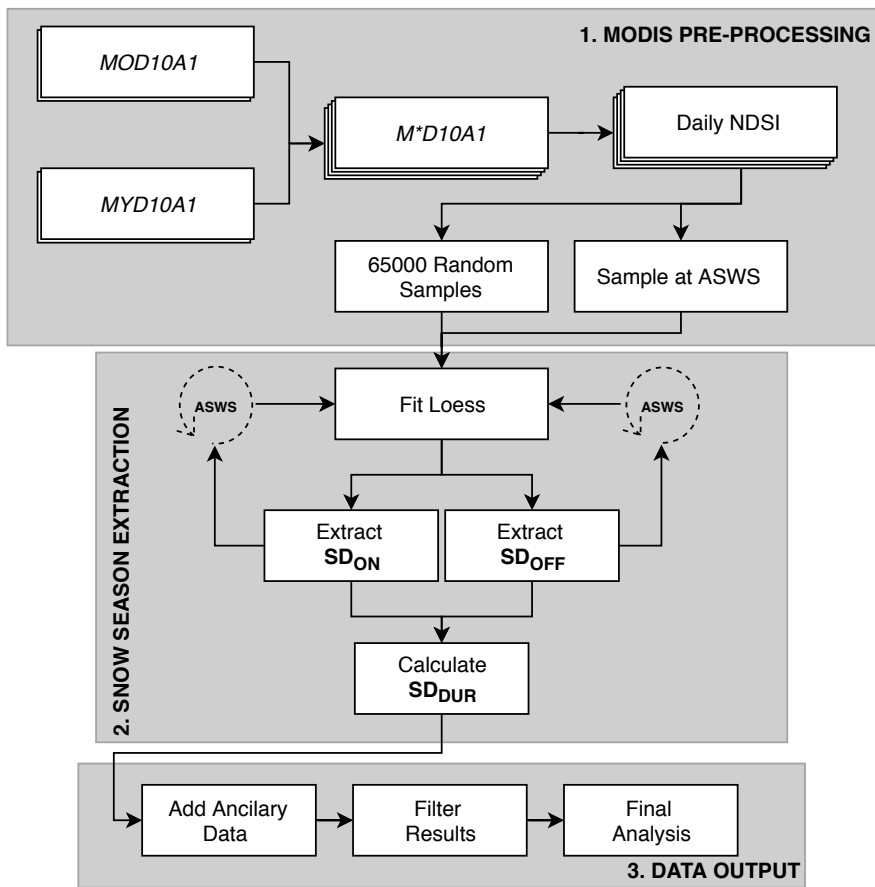

**Figure 3.** Flowchart of methods used in this study: 1) MODIS pre-processing; 2) snow season extraction; and 3) data output.

removed from the time series creating gaps of missing NDSI data. Our method, explained in the following sections, interpolates the missing data in the time series. We apply a threshold minimum threshold of 50 valid MODIS observations.

## 4.2 Snow season extraction

This study relies on a novel LOWESS temporal interpolation of the daily maximum NDSI to determine $SD_{ON}$, $SD_{OFF}$ and

5  therefore $SD_{DUR}$ of the longest snow cover period during the hydrologic year. A LOWESS interpolation is a locally weighted regression technique which permits the interpolation of missing data without over smoothing, which would result from fitting a global function. The LOWESS proceeds by estimating a smoothed value ($y_i^S$) by taking a specified bandwidth, or search window, between 0–1, that represents a fraction of the total values of $x$ to define the $N$ closest observations and estimate $y_i^S$ using a weighted (tricube in this case) linear regression (Cleveland, 1979). In order to determine the dates of $SD_{ON}$ and $SD_{OFF}$

10  at a location, the smoothed time series $y^S$ was used implementing a NDSI threshold to classify a given pixel as snow covered.





The LOWESS was fit using the Python open *Statsmodels* package. For a given hydrologic year, if there were intermittent snow covered periods, the longest continuous snow covered period was used to define $SD_{DUR}$ for that season.

### 4.3 Data output

A 500 m gridded raster of BC represents ~3.8 million MODIS pixels. Our 65 thousand samples represent a density of ~1.7 samples per 10 $km^2$. We only used samples identified as land that had more than 50 observations (days) per year which reduced our sample size to 51,276 samples. Additionally, we used SRTM to assign the elevation of each sampled location.

### 4.4 Workflow calibration

We iterated the NDSI threshold and LOWESS bandwidth values to minimize the mean absolute error (MAE) between 75% of the ASWS stations and the corresponding MODIS values for $SD_{ON}$ and $SD_{OFF}$. We used NDSI thresholds of 10–50 at intervals of 5, for both $SD_{ON}$ and $SD_{OFF}$, and LOWESS bandwidth values of 0.1–0.4 at increments of 0.05, all combinations were iterated over all of the ASWS stations. The MAE was calculated for $SD_{ON}$ and $SD_{OFF}$ and added to provide a combined MAE metric. Due to the discrepancy in the scale of the two SD observations (i.e., *in situ* vs. M*D10A1) there often exists a systematic measurement bias between the MODIS and ASWS datasets. For each parameter combination, the median difference was determined per ASWS station and used to correct data for bias present in $SD_{ON}$ and $SD_{OFF}$ at each of the stations. The median bias corrected data were used to calculate the final combined MAE. The combination of parameters that minimized the bias-corrected combined MAE were implemented in determining the final $SD_{DUR}$ from the M*D10A1 data. If parameter sets were very close in combined MAE, the standard deviation of the residual distributions was also considered for $SD_{ON}$ and $SD_{OFF}$. Then we validate the optimal thresholds against the remaining 25% of the ASWS stations to assess the transferability of our findings. Future work could include a "per station" analysis of slope, aspect, land cover and other location attributes.

### 4.5 Statistical analysis and error estimation

In this study, we use linear least squares (LLS) regression to test the effect of our explanatory variables (ONI, PDO, time and elevation) on the response variables ($SD_{ON}$, $SD_{OFF}$ and $SD_{DUR}$). In addition, we apply the non-parametric Spearman rank-order correlation ($r_S$) to determine the relative strength of these relationships. We intentionally imply causation between both the ONI and PDO on the measurements of $SD_{ON}$, $SD_{OFF}$ and $SD_{DUR}$ and seek to test that relationship. We support our assumption of causation with a corresponding theoretical framework throughout our discussion. The $r_S$ was used because it does not require any distributional assumptions. We chose to use both of these statistical tests (LLS and $r_S$) as they each have advantages and disadvantages. LLS provides a tangible unit of days per degree Celsius anomaly (d $°C^{-1}$) that is easily interpreted and useful in hydroclimatological applications, however, the LLS results are influenced by the magnitude of the explanatory variable. The $r_S$ is a well suited metric to determine correlations with the multiple explanatory variables, but cannot easily be applied to predict hydroclimatology.

Using 10 thousand iterations of our workflow, we adjust the values of $SD_{ON}$, $SD_{OFF}$, and $SD_{DUR}$ with randomly selected values



from the actual difference between the MODIS and ASWS data for each measurement to assess the impact of the inherent measurement uncertainty on our results. The error bars presented throughout this study were determined from the interquartile range of the resulting distribution of the iterated statistic.

## 5 Results

### 5.1 Workflow validation

Comparisons of the M*D10A1 derived $SD_{DUR}$ and $SD_{DUR-ASWS}$ were made over the study period (Fig. 4). The spatially corresponding M*D10A1 pixel was compared for each ASWS station and for each hydrologic year and we calibrated our workflow using 75% of the ASWS data. The parameter values that minimized the combined MAE were a LOWESS bandwidth of 0.2, and a NDSI threshold of 30 for $SD_{ON}$ and $SD_{OFF}$. The error distributions for $SD_{ON}$, $SD_{OFF}$ and $SD_{DUR}$ are summarized in Fig. 4. The bias corrected MAE of 8.7, 8.9 and 13.1 days was observed for $SD_{ON}$, $SD_{OFF}$ and $SD_{DUR}$, respectively. We acknowledge that other bandwidth and NDSI threshold combinations may improve results in other geographic regions or within sub-regions of BC, and is dependent on the training data used in this study. However, we caution over optimization of parameters due to the inherent difference is scale between the MODIS pixels and the ASWS snow pillows. Using the remaining 25% of the ASWS stations, which we use as a validation dataset, we find bias corrected MAE of 12.7, 12.6 and 16.6 days for $SD_{ON}$, $SD_{OFF}$ and $SD_{DUR}$, respectively (Fig. 4).

### 5.2 Snow Water Equivalent

We define the presence of snow from SWE measurements using a threshold of 20 mm. We chose a threshold of 20 mm by visually inspecting a subset of ASWS stations, other work as applied a threshold of 0 mm (e.g., Lindsay et al., 2015) and 5 mm (e.g., Marks et al., 2013). We then identified the longest continuous duration of SWE over the hydrological year (1 Sep–1 Sep) and record the start ($SD_{ON-ASWS}$), end ($SD_{OFF-ASWS}$) and duration ($SD_{DUR-ASWS}$) of snow cover on a station by station basis. Although M*D10A1 does not directly provide an indication of SWE, using the ASWS SWE time series we see $SD_{DUR-ASWS}$ exhibits a significant exponential relationship with corresponding peak SWE (Fig. 5). The observed relationship between peak SWE and $SD_{DUR-ASWS}$ suggests the findings of this study are relevant to water managers and seasonal forecasters.

### 5.3 Snow season results

We summarize MODIS derived snow season results in Fig. 6 as deviations from the zonal mean for each hydrozone, and for all BC. These values are not for the entire zones, rather only for areas that meet the minimum observation criteria. The hydrozone with the longest mean annual $SD_{DUR}$ is the Northern Coast Mountains (252 days), likely because of its relatively high elevation, latitude and proximity to the Pacific Ocean, leading to ample snow accumulation and glaciation. The hydrozone with the shortest $SD_{DUR}$ is Haida Gwaii (70 days), likely because of relatively low elevation, and a substantial maritime influence causing moderate winter temperatures and winter precipitation as rain. When comparing interanual variability, the





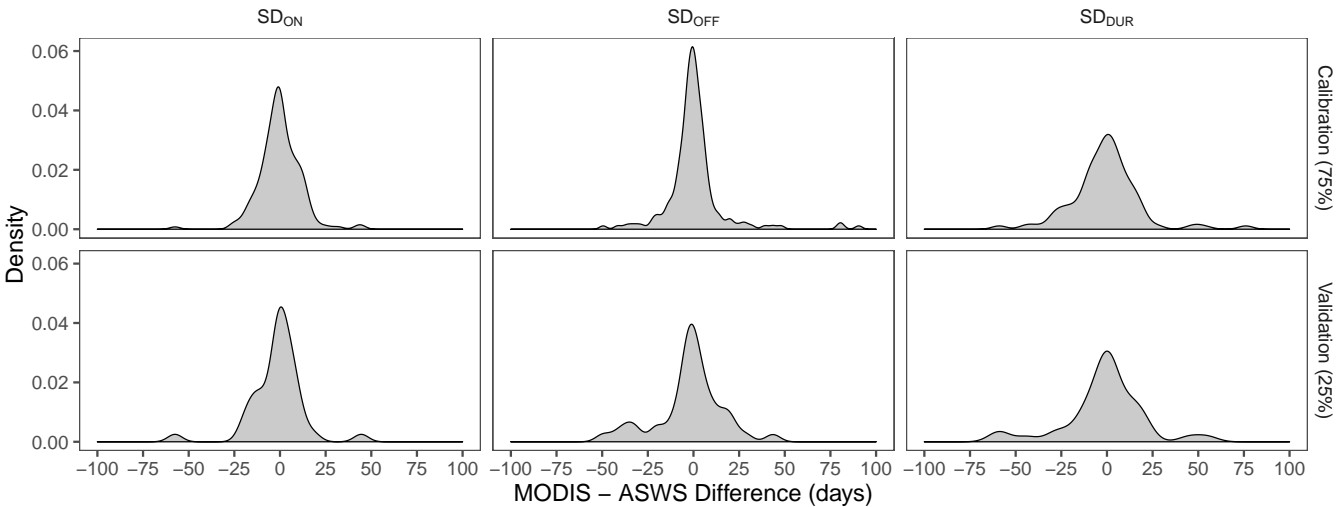

**Figure 4.** Distribution of errors calculated from the difference between MODIS and ASWS values for each snow measurement. Using the calibration dataset (75% of the ASWS data), for $SD_{ON}$, $SD_{OFF}$ and $SD_{DUR}$, respectively, the mean error is 0.82, 0.62, and -1.64 days; and the mean absolute error (MAE) is 8.7, 8.9 and 13.1 days. Using the validation dataset (25% of the ASWS data), for $SD_{ON}$, $SD_{OFF}$ and $SD_{DUR}$, respectively, the mean error is 2.5, -2.1, and -4.0 days; and the MAE is 12.7, 12.6 and 16.6 days.

following patterns emerge: $SD_{DUR}$ is anonymously short over the 2002, 2004, 2014, and 2015 hydrological years with mean provincial $SD_{DUR}$ anomalies of -13, -9, -13 and -19 days, respectively. Conversely, anonymously long $SD_{DUR}$ years include the 2006, 2007, 2010, and 2011 hydrological years with mean provincial $SD_{DUR}$ anomalies of +10, +16, +12 and +13 days, respectively (Fig. 6). However, there is much inter-zonal variability in SD for any given year. Interestingly, some short snow
5    seasons are caused by positive $SD_{ON}$ anomalies (e.g., 2002), and others are caused by earlier $SD_{OFF}$ anomalies (e.g., 2004, 2014, 2015). Using (Fig. 2), we do not interpret the positive difference from the zonal mean in 2002 as an error, this event is likely the result of a combined strong ONI and PDO.

### 5.4 Ocean–climate teleconnections

#### 5.4.1 Provincial results

10    Provincially, LLS and $r_S$, for both ONI and PDO, have significant negative relationships with $SD_{OFF}$ and $SD_{DUR}$, and no significant relationships were found with $SD_{ON}$ (Table 1). ONI has a larger magnitude LLS coefficient than PDO for both $SD_{OFF}$ and $SD_{DUR}$, however the $r_S$ results are roughly the same indicating a similar degree of correlation. The largest magnitude effect is between the ONI and $SD_{DUR}$ with a coefficient of $-12.27 \pm 4.45$ d $°C^{-1}$ (p < 0.001). The strongest negative correlation is between ONI and $SD_{DUR}$ ($-0.81 \pm 0.46$, p < 0.001). We conclude that, similar to previous studies (e.g., Fleming and Whitfield,




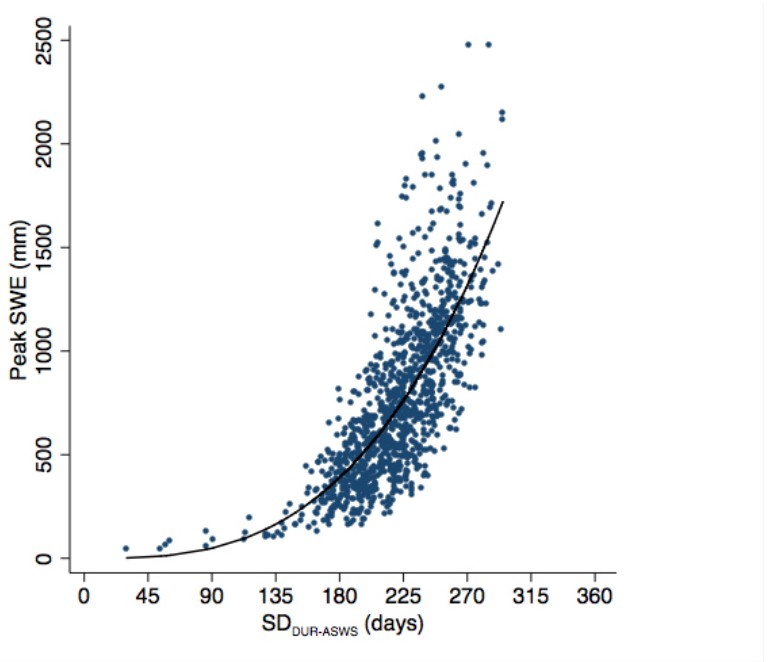

**Figure 5.** Scatter plot of $SD_{DUR-ASWS}$ and the corresponding peak snow water equivalent (SWE) for all automatic snow weather stations (ASWS) stations with valid data for a given year. The general relationship between $SD_{DUR-ASWS}$ and peak SWE appears non-linear and described best by a power function.

**Table 1.** Provincial summary of a linear least squares regression (LLS) in days per teleconnection degree Celsius anomaly (d $°C^{-1}$) and Spearman rank–order correlations ($r_S$) for $SD_{ON}$, $SD_{OFF}$ and $SD_{DUR}$ with ONI and PDO. Coefficient significance is given by the number of stars next to a value. Teleconnection values are the annual mean of the four seasonal values (Fig. 2).

| Measurement | Teleconnection | LLS (d $°C^{-1}$) | $r_S$ |
|---|---|---|---|
| $SD_{ON}$ | ONI | 1.65±2.85 | 0.02±0.21 |
| | PDO | 0.99±2.25 | 0.16±0.28 |
| $SD_{OFF}$ | ONI | -10.61±3.27*** | -0.80±0.33*** |
| | PDO | -7.55±2.55** | -0.75±0.29** |
| $SD_{DUR}$ | ONI | -12.27±4.45*** | -0.81±0.46*** |
| | PDO | -8.54±3.34** | -0.78±0.44*** |

Significance level: * = p < 0.05, ** = p < 0.01, *** = p < 0.001

2010; Moore and McKendry, 1996; Whitfield et al., 2010), El Niño (La Niña) events and Positive (Negative) PDO events result in earlier (later) melt and shorter (longer) snow duration in BC.





When we compare seasonal ONI and PDO values to provincial snow measurements ($SD_{ON}$, $SD_{OFF}$ and $SD_{DUR}$), we see seasonal variation in the response (Fig. 7). The two largest magnitude LLS relationships exists between both $SD_{DUR}$ ($-13.31\pm5.11$ d $°C^{-1}$, $p < 0.005$) and $SD_{OFF}$ ($-11.33\pm3.81$ d $°C^{-1}$, $p < 0.005$) with the Summer ONI (Fig. 7). During the period of study, moderate to strong ENSO phases crossed the $+0.5/-0.5$ $°C$ SST threshold during the summer and in some cases persisted

through until the following summer, whereas weaker events did not follow this pattern. Although the results are similar between the LLS and the $r_S$ methods, the $r_S$ is most appropriate for comparing between the two teleconnections. The two strongest $r_S$ coefficients exist between the Winter PDO for both $SD_{DUR}$ ($-0.86\pm0.50$, $p < 0.001$) and $SD_{OFF}$ ($-0.79\pm0.34$, $p < 0.001$). In general, we find that the Fall is the second highest correlated season after the Winter. There are no significant $r_S$ correlations between $SD_{ON}$ and any of the seasons, nor are there any significant $r_S$ correlations between the Spring teleconnections and any

of the snow measurements (Fig. 7).

### 5.4.2   Regional patterns

Important regional patterns emerge when we aggregate the data by hydrozone. These patterns broadly reflect the continentality, latitude and elevation of the hydrozones. Very few hydrozones outside of northwest BC have significant LLS relationships between any of the seasonal teleconnections and $SD_{ON}$ (Fig. 8). Of those that are significant, the three greatest magnitude

LLS for $SD_{ON}$ are all with the Spring ONI, they are Stikine Plateau ($6.08\pm4.01$ d $°C^{-1}$, $p < 0.05$), Northern Coast Mountains ($5.24\pm3.85$ d $°C^{-1}$, $p < 0.05$) and Northern Central Uplands ($4.57\pm3.89$ d $°C^{-1}$, $p < 0.05$). For $SD_{OFF}$, most of the province has significantly negative LLS with the Summer, Fall and Winter ONI values (Fig. 8). The three strongest relationships with $SD_{OFF}$ are found in Eastern Vancouver Island ($-18.68\pm3.86$ d $°C^{-1}$, $p < 0.005$), Central Coast Mountains ($-17.60\pm3.63$ d $°C^{-1}$, $p < 0.005$), and Western Vancouver Island ($-17.30\pm3.80$ d $°C^{-1}$, $p < 0.05$), all with Summer ONI. For $SD_{DUR}$, we find

the greatest magnitude trends, and a spatial distribution that closely resembles $SD_{OFF}$. The three strongest relationships with $SD_{DUR}$ are found in Western Vancouver Island ($-24.82\pm5.13$ d $°C^{-1}$, $p < 0.01$), Eastern Vancouver Island ($-22.87\pm5.01$ d $°C^{-1}$, $p < 0.01$) and the Central Coast Mountains ($-21.35\pm4.93$ d $°C^{-1}$, $p < 0.01$) with the Summer ONI. All values for all regions are reported in the Supplementary Materials Table 7.

Using $r_S$ coefficients we compare spatial patterns in the relationship of the teleconnections with SD. Similar to the LLS results,

the $r_S$ correlations for $SD_{ON}$ are only significant for a few hydrozones. Positive correlations occur in northwest BC during the Spring ONI and the Fall and Winter PDO (Fig. 8). The three strongest correlations with $SD_{ON}$ are Northern Coast Mountains with the Winter PDO ($0.64\pm0.65$, $p < 0.01$), Western Vancouver Island with the Summer ONI ($0.63\pm0.40$, $p < 0.01$) and Northern Coast Mountains with the Fall PDO ($0.61\pm0.62$, $p < 0.05$). For $SD_{OFF}$, we find similar significance patterns as those found for LLS. Namely, the effect of Spring teleconnections on $SD_{OFF}$ in the southern half of the province is generally not

significant, and when there exists a significant effect, it is always negative. All significant $SD_{OFF}$ correlations fall between $-0.5$ and $-0.91$. The strongest correlations are found in Okanagan Highland with the Winter ONI ($-0.91\pm0.36$, $p < 0.001$) and the Fall ONI ($-0.89\pm0.36$, $p < 0.001$), and in Southern Thompson Plateau with Fall ONI ($-0.87\pm0.35$, $p < 0.001$). As for the $SD_{DUR}$, we again find good agreement with LLS. Few regions display significance in the northeast, and the results are negative throughout the province. There are few regions that have significant correlations in southern BC with either the Spring



teleconnections or with the summer PDO (Fig. 8). The strongest relationships are found in Southern Quesnel Highland with the Winter ONI (-0.90±0.41, p < 0.001), the Fall ONI (-0.87±0.39, p < 0.001) and the Northern Columbia Mountains with the Fall ONI (-0.86±0.41, p < 0.001).

## 5.5 Elevation dependency

Using $r_S$ and LLS (Fig. 9) we find that for $SD_{ON}$, all significant results (p < 0.05) have positive median $r_S$ and LLS (Fig. 9). Indicating that when the ONI and/or PDO are in their warm phase, $SD_{ON}$ increases (becomes later) to a similar degree at all elevations. The range of values per elevation bin is smaller for the ONI than the PDO at low elevations. However the number of significant relationships diminishes above 2000 m asl. Nearly all $SD_{OFF}$ $r_S$ and LLS values are negative for all elevations and trend towards zero at 2500 m asl. Below 2500 m asl, values decrease until 500 m asl, and increases again at 0. This suggests
that lower elevations are more sensitive to changes in ONI and PDO. $SD_{DUR}$ has a very similar distribution by elevation as $SD_{OFF}$. These results provide evidence that there are significant interactions between elevation and the ONI and PDO influence on $SD_{OFF}$ and $SD_{DUR}$ regionally over BC, with the largest magnitude and most highly correlated relationships occuring at lower elevations.

## 5.6 Change over time

Provincially, the annual MODIS snow cover metrics have insignificant negative trends over time. $SD_{ON}$ has the largest magnitude LLS result -0.29±0.18 d yr$^{-1}$ (p > 0.1) with a very low $r^2$ value of 0.08. $SD_{OFF}$ also has a statistically insignificant negative LLS results of -0.23±0.36 d yr$^{-1}$ ($r^2$ = 0.01, p = 0.53) and $SD_{DUR}$ has a positive LLS result of 0.07±0.40 d yr$^{-1}$ ($r^2$ = 0.0009, p = 0.87). When we detrend the time series using the annual linear relationships with ONI and PDO found earlier, the three measurements still show insignificant trends over time (see Supplementary Materials Table 13). Further province-wide
analysis by season indicates the same result of insignificant trends over time for both the original and the detrended data (see Supplementary Materials Table 14).

We look at change over time for each elevation band by hydrozone (Fig. 10). For $SD_{ON}$, a few regions throughout the Province display negative trends over time, for example, $SD_{ON}$ in the Southern Rocky Mountain Foothills in the 0–500 m has LLS results of -1.45±0.63 d yr$^{-1}$ (p < 0.05) indicating a change towards earlier start dates. The northwest of the province over
elevations 1500–2000 m, however, has strong positive trends (later start dates) of $SD_{ON}$ over time that is coherent between adjacent regions (Fig. 10). For $SD_{OFF}$, most regions in northern BC have significantly negative $SD_{OFF}$ at nearly all elevations. The Northern Thompson Plateau has a strong positive trend of 1.61±0.54 d yr$^{-1}$ (p < 0.01) in $SD_{OFF}$ in the 1000–1500 m group (Fig. 10). For $SD_{DUR}$, we find similar patterns to $SD_{OFF}$, where a large portion of northern BC has significant negative trends over time and these trends are stronger towards 2000 and 2500 m and portions of the southern interior of BC below 1500 m
have positive trends (Fig. 10). The southern interior has experienced substantial land cover change caused by forestry (e.g., Hansen et al., 2013) and has experienced extensive forest health outbreaks (Boon, 2012) and wildfires (BCWS, 2018) over the MODIS time period. The Fraser Plateau, for example, experienced a total of 26,869 km$^2$ of forest fire between 1917–2017 (inclusive), and of that nearly 40 % (10,253 km$^2$) occurred in 2017 (BCWS, 2018).



When we detrend the snow measurements for each of the elevation bands in each hydrozone using a combination of the ONI and PDO relationships, we find two strong regional signals. They are both in the north and northwest of BC in the 2000 m group. $SD_{ON}$ has significant positive values for many adjacent hydrozones for each of the seasonal indices of ONI and PDO, the strongest of which is found in the North Central Uplands ($0.62\pm0.24$ d yr$^{-1}$, $p < 0.05$) and $SD_{DUR}$ has significantly negative

values in the northern BC with the strongest signal again in the North Central Uplands ($-1.76\pm0.69$ d yr$^{-1}$, $p < 0.05$). All results are available in the Supplementary Materials Table 17.

## 6 Uncertainties

There are a number of uncertainties associated with this study. These uncertainties relate to the: 1) short time period of the

MODIS record; 2) missing data due to cloud cover and shadows; 3) using ASWS to calibrate and validate our method; 4) land cover changes; 5) hydrozone bias; and 6) a limited number of ONI and PDO events occurring during the study period.

1) The short time period of the MODIS record is commonly used to study incremental change such as greening (e.g., Zhang et al., 2017). However, using M*D10A1 to detect change over time is perhaps overstepping dataset due to the high inter-annual variability of snow cover. Nevertheless, many have shown that snow cover has in fact significantly changed over the MODIS

period of record (e.g., Li et al., 2018; Lindsay et al., 2015; Saavedra et al., 2018).

2) Overcoming missing data due to clouds and shadows in the MODIS record is difficult and many have demonstrated methods to overcome this shortcoming (e.g., Krajčí et al., 2014; Marchane et al., 2015), we have demonstrated a LOWESS to interpolate missing daily NDSI data.

3) Our validation dataset of 60 ASWS stations throughout BC is very strong in that we can have a large sample of validation

points. However, these snow pillows are typically near treeline and are concentrated in southern BC. This limitation implies that our calibration and validation steps are biased to work best at treeline in southern BC (Fig. 1). The lack of systematic snow cover data elsewhere that covers a suitable range of elevations limits us from constraining this uncertainty further.

4) BC has undergone significant landscape changes over the last twenty years. These include widespread forest insect outbreaks (e.g., Stahl et al., 2006b), severe wildfire seasons (e.g., Crowley et al., 2019), summer drought (e.g., Barnett et al., 2005), and

extensive harvesting of forests (e.g., Hansen et al., 2013). All of these land cover changes have important effects on snow hydrology (Parajka et al., 2012), and may act to either enhance or reduce the observed interannual SD response to both the ONI and PDO. Forest canopies are known to impede the detection of snow on the ground from satellite remote sensing (e.g., Boon, 2012; Rittger et al., 2013), however, the effect of land cover type on the optimal parameter values in this study, are unknown.

5) We summarize our results using hydrozones in order to capture much of the variability in SD over BC (Coulson and Obedkoff, 1998). There are, however, other logical ways to group the province (e.g., 1 degree latitude and longitude groups, ecoprovinces, ecozones, etc.). The effect of using hydrozones instead of another zonal dataset on our results is unknown.

6) The PDO index is likely influenced by moderate to strong ENSO events enhancing SST's in the north Pacific Ocean. The





effect of combined PDO and ONI indices on results was beyond the scope of this paper. The PDO index during the period of study was also influenced by a marine heat wave commonly referred to as "The Blob" (Walsh et al., 2018). This unique event occurred 2013–2015 and led to warm and dry conditions over much of the study area, which may have influenced results during that time.

## 7    Discussion

### 7.1    Snow season extraction

The LOWESS interpolation proved to be a robust method for determining $SD_{ON}$, $SD_{OFF}$ and $dSD_{DUR}$ of the longest continuous snow covered period from the daily M*D10A1 data. Combining the daily data from both the Terra and Aqua platforms helped to maximize the temporal resolution of our estimation of $SD_{ON}$ and $SD_{OFF}$. Calibrating our workflow to a subset of the ASWS resulted in MAE values slightly lower than those reported in other studies (e.g., Lindsay et al., 2015).

Figure 6 summarizes the snow cover results per hydrozone, and for all BC. We urge caution with their interpretation, as there are a number of limitations to our methodology, for example, Haida Gwaii, as a whole, does not experience a snow season of 70 days, however, we've found that for the areas with seasonal snow cover, that is the average duration of snow. Multiple spatio–temporal patterns can be observed in Fig. 6, however, this study focuses primarily on the regional patterns in the influence of ocean–climate teleconnections on the timing and duration of snow cover.

### 7.2    Regional patterns

We summarize $r_S$ results to identify the strongest significant correlation between the $SD_{ON}$, $SD_{OFF}$ and $SD_{DUR}$ and the teleconnections by season (Fig. 11) and annually (Fig. 12). Observing $SD_{ON}$, we see that regions with significant results are located in northwest BC, and sporadically in the southwest (Fig. 11). $SD_{OFF}$ has important regional patterns when we highlight the strongest Spearman correlations, notably that southern BC has a matrix of Fall and Winter for both ONI and PDO, with some exceptions in the higher elevation hydrozones being sensitive to the Summer ONI. Differences include northeastern BC, which has no significant relationships with ONI and yet has a coherent signal with the Summer PDO. Northwestern BC overall is more sensitive to the Spring and Summer ONI, and to the Fall PDO. $SD_{DUR}$ closely resembles the $SD_{OFF}$ with some differences. The southern half of BC reveals a matrix of ONI influences from the Summer, Fall and Winter ONI, the northwest is sensitive to the Spring ONI. As for PDO, a longitudinal transect exists with the Fall PDO dominating the coastal hydrozones and the Winter dominating inland. Northeastern BC is once again not well correlated with the teleconnections.

We compare mean annual correlations to determine which teleconnection explains more variation in SD regionally (Fig. 12). We find that $SD_{ON}$ is largely not well correlated with teleconnections, however in the northwest, the PDO is most important. For $SD_{OFF}$ we see strong regional patterns where northern and southern BC are most highly correlated with the PDO and ONI, respectively. As for $SD_{DUR}$, the pattern is almost identical to $SD_{OFF}$ with a few exceptions.





The ONI and PDO are more correlated with $SD_{OFF}$ than $SD_{ON}$ regardless of season (Fig. 8, Fig. 11, Fig. 12). This is likely because precipitation and temperature anomalies associated with ONI and PDO are poorly established early in the snow season (Shabbar and Khandekar, 1996). Bonsal et al. (2001) demonstrate that the centre of winter temperature anomalies in response to ENSO moves from southwest Yukon into northern and central Alberta, to a centre over Alberta during the PDO neutral

phase, to neutral or even negative temperature anomalies when the PDO is in its cool phase partially explaining the north-south gradient in the amount of SD variation over BC (Fig. 12).

$SD_{OFF}$ is highly correlated with either the Fall or Winter ONI in southern BC, however, in the Stikine Plateau and Northern Rocky Mountains Spring ONI explains more variation in $SD_{OFF}$, and Summer ONI explains more variation in the Northern Coast Mountains and Northern Central Uplands. We hypothesize these correlations between $SD_{OFF}$ and the Spring and Sum-

mer ONI may be present in northwest BC because they occur near the regional origin (northwest BC) of the temperature anomalies that manifest in response to a El Niño event. The strong correlations between $SD_{OFF}$ over most of southern BC and the Fall and Winter ONI are consistent with the corresponding winter temperature and precipitation anomalies over Western Canada observed in other studies (e.g., Bonsal et al., 2001; Shabbar, 2006; Stahl et al., 2006a). The results indicate that $SD_{OFF}$ within both the McGregor Basin and the Upper Fraser Basin are more highly correlated with the PDO than the ONI, despite

being located in the centre of the documented winter temperature and precipitation anomalies that occur in response to El Niño (Shabbar, 2006), this may indicate a likely interaction between the ONI and PDO in these hydrozones.

Elevation bins of 500 m for each hydrozone dramatically increases the amount of regions with significant results, particularly for $SD_{ON}$ (Fig. 9). Changes of LLS and $r_S$ with elevation are not particularly apparent with $SD_{ON}$, however important differences exist for $SD_{OFF}$ and $SD_{DUR}$. High elevation snow cover timing and duration appear more resilient to ONI and PDO

(values trend towards 0 at treeline).

Ocean–climate teleconnections may be influenced by anthropogenic climate change (Cai et al., 2018; Wittenberg, 2009). The interplay between naturally present variability in snow duration and timing and the variability introduced by anthropogenic forces is difficult to untangle with such a short period of record. This is exacerbated by the generally poorly understood mechanisms of longer lived climate modes such as the PDO (Rodenhuis et al., 2007). One study suggests that increases in Equatorial

Pacific El Niño SST variance are related to stratification of the upper equatorial Pacific Ocean under greenhouse warming, leading to enhanced wind–ocean coupling conducive to SST anomalies (Cai et al., 2018). Other studies suggest that there is an insufficient period of record to determine the effects of climate change on ENSO given strong interdecadal and intercentennial modulation of its ENSO behavior in controlled simulations (e.g., Wittenberg, 2009).

## 8   Conclusions

The main objective of this study was to evaluate the suitability of using daily M*D10A1 imagery to understand the regional influence of ocean–climate teleconnections on $SD_{ON}$, $SD_{OFF}$ and $SD_{DUR}$ in BC, a region of nearly 1 million km$^2$ with a low density of snow observations. We have used a new method of fitting a LOWESS interpolation to the daily M*D10A1 NDSI time series. This method helps account for clouds and missing data while maximizing the temporal resolution when compared





to 8 or 16 day snow products that are frequently used. We confirm the general consensus from many previous studies that both ENSO and PDO have significant impacts on the timing and duration of seasonal snow cover in BC. We add to this knowledge with seasonal and regional analysis using the hydrozones of BC and elevation bins of 500 m.

We found that the optimal NDSI threshold values in our study region was 30 when compared with ASWS SWE data. We
separate automatic snow weather station data into calibration (75%) and validation (25%) subsets and obtain mean absolute errors between the MODIS and *in situ* observations for the start, end and duration of 8.7, 8.9 and 13.1 days for the calibration data, and 12.7, 12.6 and 16.6 for the validation data, respectively.

$SD_{ON}$ generally has insignificant or weak correlations with ONI and PDO in BC, with the exception of northwest BC that has a generally positive linear relationship with PDO, and the Fraser Plateau that is significantly negatively correlated with PDO
(Fig. 8). $SD_{OFF}$ has significantly negative correlations with both teleconnections throughout BC, however, the Winter and Fall PDO are the most highly correlated seasonal components. By hydrozone, the effects of both PDO and ONI are greatest in the southwest and dissipate towards northeastern BC (Fig. 8). The strongest seasonal ONI correlation varies throughout the province with the Winter dominating the interior, whereas the strongest seasonal PDO correlation is the Winter in southern BC, and the north is split by the Summer and Fall in northeastern and northwestern BC, respectively (Fig. 8). Overall, ONI is more
highly correlated in southern BC and PDO northern BC (Fig. 12). The greatest magnitude effects are at lower elevations, and they trend towards neutral effects at higher elevations. $SD_{DUR}$, similarly to $SD_{OFF}$, has a significantly negative relationships with both teleconnections throughout BC.

We observe two strong signals of change over time, generally declining snow cover at higher elevations in northwest BC, and generally increasing snow seasons at lower elevations in the southern interior of BC. These patterns, however, change when
we detrend our data using linear relationships with ONI and PDO. Once detrended, only two regions in northwest BC show a significantly declining snow cover over time. The most important rate of change over time in the detrended $SD_{DUR}$ is found in the North Central Uplands (-1.78±0.67 d yr$^{-1}$, p < 0.05) accounting for nearly a month of change over the MODIS period of record. The non-detrended positive change over time in the southern interior is likely due to important land cover change over that time. Forestry, forest health and wildfire have all been very present on the landscape. We interpret this as one of the
challenges of using a 16 year time series to understand snow cover change over time.

Future directions for this research could include the creation of new snow zones for BC, the investigation of seasonal snow predictions, stream runoff, peak SWE, and otherwise getting closer to a solid approximation of SWE from historical remote sensing datasets. Also a robust investigation of the influence of land cover change and the influence of terrain (slope, aspect and curvature) on MODIS derived snow cover would be helpful. Visible Infrared Imaging Radiometer Suite (VIIRS) sensor aboard
the Suomi NPP (launched 2011) and the NOAA-20 (launched 2017) satellite platforms provide excellent future opportunities to continue snow monitoring using similar temporal and spatial resolution imagery. Other avenues of investigation include the high temporal frequency and spatial resolution PlanetScope Dove constellation with ∼1-2 day repeat, ∼3 m resolution (Planet Team, 2017). Although NDSI cannot be calculated from PlanetScope data as there is no SWIR band, other indices, such as NDVI may be used to approximate snow cover.



*Code availability.* https://github.com/bcgov/ts-rs-modis-snow

*Data availability.* Data will be made available on the British Columbia Data Catalogue (https://data.gov.bc.ca/) following peer-review.

*Author contributions.* Bevington and Gleason conducted the majority of the analysis and drafting of the manuscript. Foord contributed mainly to interpretation of the results. Floyd and Griesbauer contributed to the high level direction and interpretation of the project as a whole.

*Competing interests.* We have no competing interests to declare.

*Acknowledgements.* The authors are very grateful to S. Déry, M. Allchin, Y. Wang, J. Rex and W. Mackenzie for conversation and enthusiasm that helped inform the early direction of this work. Also, we acknowledge the open data science community that makes this work possible, specifically free access to MODIS, Google Earth Engine, R-statistical packages, Python and Statsmodels, and QGIS.





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





**Figure 6.** Difference from the mean of SD$_{ON}$, SD$_{OFF}$, and SD$_{DUR}$ (in days) for each hydrozone and for all BC, over time. Mean zonal values (in days after 1 Sep for SD$_{ON}$ and SD$_{OFF}$, and in days for SD$_{DUR}$) are found at the right hand side of each plot. The x-axis is in hydrological years starting on 1 Sep for each year, for example, the label 2002-03 represents 1 Sep 2002–1 Sep 2003.


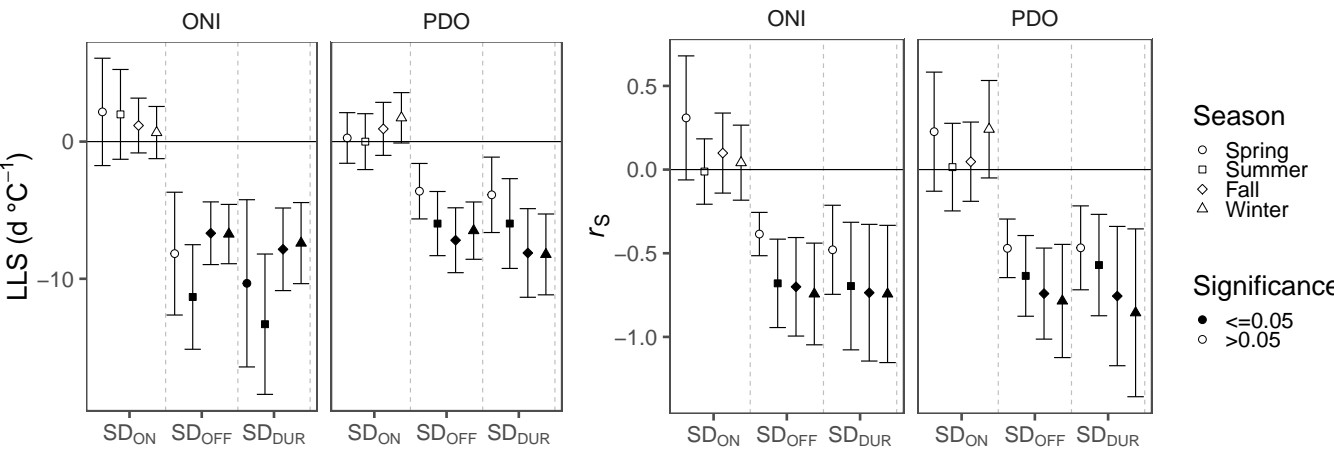

**Figure 7.** Summary of the linear least squares (LLS; left two panels) and Spearman correlation coefficients ($r_S$; right two panels) for each of the three measurements ($SD_{ON}$ $SD_{OFF}$, and $SD_{DUR}$), by teleconnection (ONI and PDO) and season. The y-axis shows the rate of change in days per degree Celsius teleconnection anomaly (d $°C^{-1}$) and the Spearman correlation coefficient ($r_S$). The point fill denotes insignificant relationships (p > 0.05) as white and significant ones (p $\leq$ 0.05) as black. Error bars represent the 25th and 75th percentiles of the ten thousand iterations where the snow measurements are adjusted using a random sample of the difference between the MODIS data and the ASWS data. See Supplementary Materials for all plotted values.



**Figure 8.** Map of BC showing the linear least squares (LLS) rate of change in days per degree Celsius anomaly (d $^{\circ}$C$^{-1}$; left panel) and Spearman correlation coefficients (right panel) for SD$_{ON}$, SD$_{OFF}$ and SD$_{DUR}$ (per column) with seasonal (Spring, Summer, Fall, Winter) ONI and PDO values (per group of four rows) by hydrozone. Grey zones have insignificant trends (p > 0.05). Use Figure 1 for spatial reference. See Supplementary Materials for raw values and error bars.

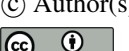

**Figure 9.** Linear least squares (LLS; top panel) and Spearman correlation coefficients ($r_S$; bottom panel) per hydrozone for $SD_{ON}$, sdoff and $SD_{DUR}$ with ONI and PDO in 500 m elevation bins. Only correlations with p-values $\leq$ 0.05 are shown. Boxplots represent the median (black horizontal line in box), 25th and 75th percentile ranges (box limits) and the 5th and 95th percentile ranges (whiskers) with outliers shown as black points. The sample size of each box is shown below it with a maximum of twenty-nine values (one per hydrozone).



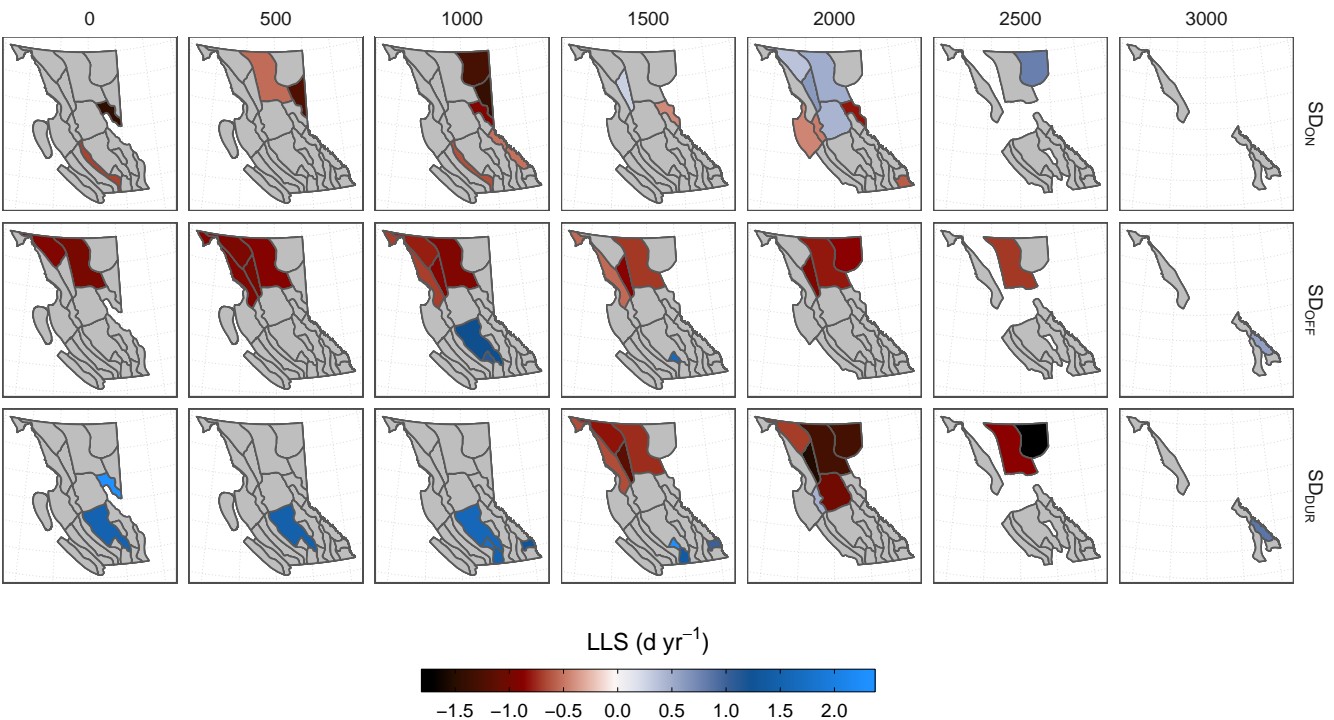

**Figure 10.** Map of British Columbia showing the rate of change in days per year (d yr⁻¹) of the MODIS derived SD$_{ON}$, SD$_{OFF}$ and SD$_{DUR}$ (vertical panel groups) by hydrozone and by elevation group of 500 m (horizontal panel groups). Grey zones have insignificant trends (p-value > 0.05), missing zones have no terrain in the given elevation bin.

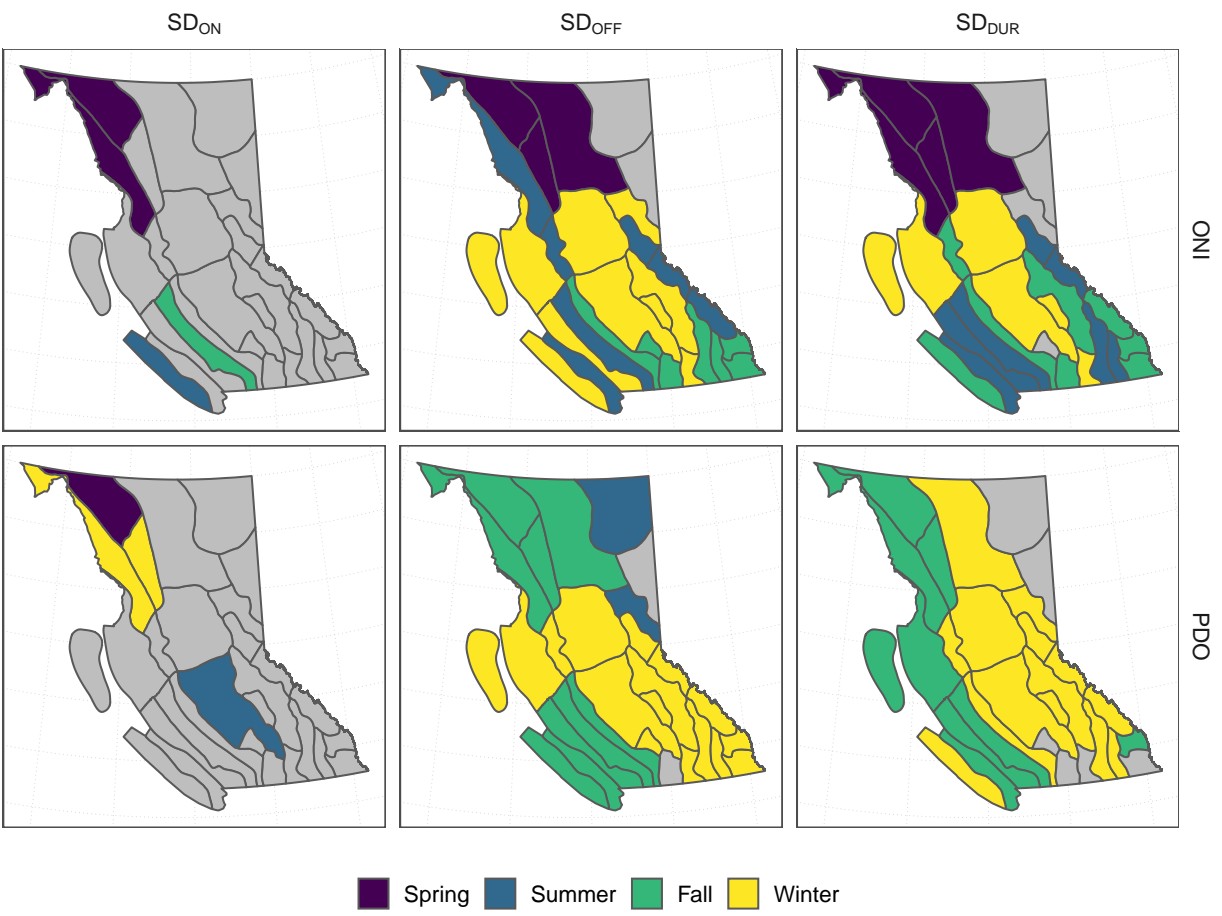

**Figure 11.** Map of British Columbia showing seasonal teleconnection index with the strongest Spearman correlation per hydrozone, for each of the four snow measurements: $SD_{ON}$, $SD_{OFF}$ and $SD_{DUR}$.
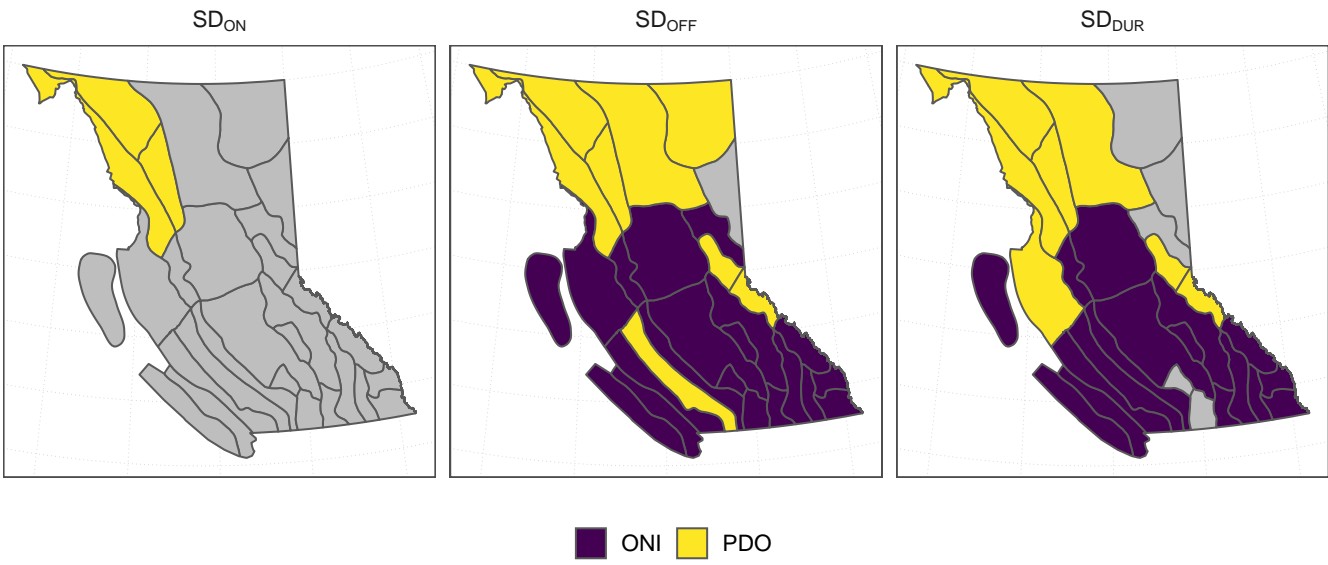

**Figure 12.** Map of British Columbia showing annual teleconnection index with the strongest significant Spearman correlation per hydrozone, for each of the three snow measurements: $SD_{ON}$, $SD_{OFF}$ and $SD_{DUR}$. Insignificant correlations ($p > 0.05$) are shown in grey.