# Peer review of "Regional influence of ocean-climate teleconnections on the timing and duration of MODIS derived snow cover in British Columbia, Canada"

_The Cryosphere, 2019_

## Referee Comment (RC1) · Anonymous Referee #1 · 8 Jul 2019

General comments

This study investigates the impact of Oceanic Niño Index (ONI) and Pacific Decadal Oscillation (PDO) on snow cover characteristics in British Columbia. The results indicate that the end duration of snow cover are strongly negatively correlated with ONI and PDO. This result is in agreement with previous studies in the region, but the novel contribution of this study is the evaluation of regional and seasonal patterns of this relation. The regional patterns show that the most correlated regions are in the northern and southern part of British Columbia.

[Figure]

Overall the study is clearly written and the topic fits well to the scope of the journal. The study is original, however my main concern is somewhat unclear/confusing formulation of the significance and novel scientific contribution of the results. As the authors clearly state the results are consistent with many previous studies evaluating the impact of teleconnections on snow characteristics in the study region. I wonder to what extent are the emerged regional patterns significant in terms of novel scientific contribution and how can these result be generalized to other regions or time periods. The novel method used for cloud filtering (by using LOWESS interpolation) is somewhat hidden in terms of formulation of the novelty. I would suggest to extend the discussion section and compare the results with some relevant previous studies (e.g. McClung, 2013, Barton (2017)) to indicate more clearly how do the new results fit to previous studies and how the interpolated (cloud reduced) MODIS datasets can improve (are improving) the prediction of the impact of ONI and PDO. Are the mean absolute errors of prediction comparable/small/large compared to results of previous investigations?

Specific comments

Section 4.1. (MODIS processing). Why not to use also MOD10A1 data from 2001 and 2002? Will it have some impact on the results? MODIS mapping accuracy. It will be interesting to see the overall accuracy of the cloud reducing (interpolation) method (based on comparison of MODIS and ASWS). This will allow to put in more context the results of error assessment of the start, end and duration of snow cover.

References

McClung Journal of Glaciology, Volume 59, Issue 216, 2013 , pp. 783-792

Mark Barton (2017) Twenty-Seven Years of Manual Fresh Snowfall Density Measurements on Whistler Mountain, British Columbia, Atmosphere-Ocean, 55:3, 144-154, DOI: 10.1080/07055900.2017.1331157
* * *

---

## Referee Comment (RC2) · Anonymous Referee #2 · 12 Jul 2019

General comments

This study investigates the regional effects of the Oceanic Niño Index (ONI) and Pacific Decadal Oscillation (PDO) on snow cover characteristics in British Columbia (BC). The strong relationships provided between snow cover and climate indices is not new but the regional analysis of changes of snow season has not been studied in detail for BC and there is a need to document these changes. The most concern is the limited length of MODIS data that cover few ONI and PDO events. However, this study provides a basis for futures studies testing longer-term relationships and the use of MODIS trend

data area common in the literature. Overall, the study is well organized and fits the scope of TC. The use of LOWESS interpolation is a novel approach but is not clear what advantages has in contrast from another cloud filled approach (e.g. Li et al, 2017; Khoramian Dariane, 2017).

Specific comments

Section 3.3. (Hydrozones)

The result shows latitudinal changes of magnitude of SDON, SDOFF, and SDDUR. Hammond et al (2018) showed a change of snow persistence in BC associated to elevation change with similar latitude and probably SD could have elevation relationship as well. I suggest running an elevation analysis to looking for elevation dependent factor in your 65 thousand locations. The elevation-latitude SD changing could be a great complement to current results.

Section 4.1. (MODIS processing)

The MODIS snow cover Collection 6 has been significantly revised and data content has been increased compared with the previous collection. For the MYD10A1 integrated a Quantitative Image Restoration (QIR) algorithm (Gladkova et al., 2012) to restore the Aqua MODIS band 6 to allow use exactly the same product for MYD10A1 and MOD10A1. I suggest including the advantages of the new collection of Aqua MODIS to highlight your novel cloud filled approach.

Section 5.1. (Workflow validation)

The NDSI threshold was 30. The previous Collection 5 had a threshold of 40 to define a pixel as snow but this fixed threshold doesn't work well in different vegetation cover condition. I suggest states the vegetation condition (as NDVI) range in your locations in order to define the limits of your NDSI threshold.

References

Gladkova, I., M. Grossberg, G. Bonev, P. Romanov and F. Shahriar, 2012: Increasing the accuracy of MODIS/Aqua snow product using quantitative image restoration technique, IEEE Geoscience and Remote Sensing Letters, 9(4):740-743. Li, X., Fu, W., Shen, H., Huang, C., and Zhang, L., 2017. Monitoring snow cover variability (2000–2014) in the Hengduan Mountains based on cloud-removed MODIS products with an adaptive spatio-temporal weighted method, Journal of Hydrology, 551, 314-327. Khoramian, A. and Dariane, A., 2017. Developing a Cloud-Reduced MODIS Surface Reflectance Product for Snow Cover Mapping in Mountainous Regions, Geosciences, 7.

---

## Author Comment (AC1) · 2 Aug 2019

Response to Anonymous Referee #1 ()

Dear Anonymous Referee #1,

Thank you for your kind and thorough review of our manuscript. Your comments were well received and we believe that we have addressed all of the issues that you have pointed out. Below, we highlight, point-by-point, how we have addressed your comments with reference to the section, page number and line numbers of the modifica-

tions.

Kind regards,

On behalf of the co-authors,

Alexandre Bevington

========================================================================

RC1 Comment 1:

"[...] my main concern is somewhat unclear/confusing formulation of the significance and novel scientific contribution of the results. As the authors clearly state the results are consistent with many previous studies evaluating the impact of teleconnections on snow characteristics in the study region. I wonder to what extent are the emerged regional patterns significant in terms of novel scientific contribution and how can these result be generalized to other regions or time periods. The novel method used for cloud filtering (by using LOWESS interpolation) is somewhat hidden in terms of formulation of the novelty. I would suggest to extend the discussion section and compare the results with some relevant previous studies (e.g. McClung, 2013, Barton 2017) to indicate more clearly how do the new results fit to previous studies and how the interpolated (cloud reduced) MODIS datasets can improve (are improving) the prediction of the impact of ONI and PDO. Are the mean absolute errors of prediction comparable/small/large compared to results of previous investigations [...]"

Response to RC1 Comment 1:

1.1

Indeed, the novel aspects of this study are somewhat hidden due to a heavier focus on the methods and results. We have added a short section titled "7.3 Significance of results" (page 17 line 32) to summarize the novel contributions of this paper. In this section we aim to highlight that we are not the first to use a LOWESS interpolation

on MODIS time series data, however, this is the first study to apply it to the MODIS snow cover product. In addition, the LOWESS allows us to detect the start and end dates of snow, which is an important contribution to our understanding of snow cover in British Columbia. We suggest that to extend this work further into the past, one could use lower resolution remote sensing data, or in-situ observations, and we include the provided references of (McClung, 2013, Barton 2017). These methods could likely be implemented in other regions, however the dates of the hydrological year will likely be different, and so could the optimal NDSI threshold. Finally, we highlight that the 500 m rasters that we have produced for this study could be of interest to other fields of science, and that our results could likely improve seasonal forecasting of snow.

Here is the new section in its entirety: (7.3 Significance of results):

We are not the first to use LOWESS time series interpolation on MODIS data (Moreno et al., 2014), however, to our knowledge this is the first study to use LOWESS on the MODIS snow cover product (Fig. 1). This allowed us to not only detect SDDUR from the time series, but also SDON and SDOFF. To extend this work further back in time, one could use lower spatial resolution AVHRR data (Allchin and Déry, 2017), or investigate in situ measurements (McClung, 2013; Barton, 2017). Our methods could be used anywhere that the MODIS snow cover product is present, however the definition of the hydrological year, the LOWESS bandwidth, and the NDSI thresholds may need to be optimized. Also, our method detects the longest period of continuous snow cover and will not be as useful for areas of sporadic snow cover.

The 500 m resolution annual SDON, SDOFF and SDDUR rasters produced as intermediate data in this study fill an important gap in our understanding of the regional influence of ocean-climate teleconnections on snow cover in British Columbia. These rasters may be useful for a number of other climatological and environmental processes in the fields of hydrology, ecology, and more. Operationally, our findings can be used to constrain seasonal forecasts of snow in British Columbia. For elevation 10 bins of 500 m by hydrozone, we have LLS and rS values for the relationships between snow cover

and seasonal ocean-climate teleconnections.

1.2

In addition, to better explain the LOWESS, and to highlight its functionality with the MODIS snow cover data, we have expanded on section 4.2 (page 8 lines 8-13) and added a figure of the LOWESS interpolation of NDSI timeseries data (Fig. 1).

Here is the new text in its entirety (4.2 Snow season extraction; page 8 lines 8-13):

LOWESS time series interpolation has been shown to be more resistent to gaps and outliers than other similar methods in a study of the MODIS fAPAR (fraction of absorbed photosynthetic active radiation) product (Moreno et al., 2014). Other studies have proposed methods of removing clouds from MODIS snow cover data. Spatio-temporal filtering that uses a temporal probability of snow and a DEM (Li et al., 2017), and a spatial k-means interpolation with dynamic time warping (Khoramian and Dariane, 2017) are some of the innovative methods being developed in the rapidly evolving field. Our focus was to find a method that easily detects SDON and SDOFF, and the LOWESS does that quickly, efficiently and accurately.

1.3

As for how our errors compare to other studies, we have highlighted specific error bars from other studies in the introduction (page 2 line 23) and, stated our errors in section 5.1 and Fig 2, then again in 7.1 (page 16 line 14).

Here is the new text in its entirety (1 Introduction; page 2 line 23):

Lindsay et al. (2015) were able to demonstrate using MODIS that nearly half of Alaska and surrounding areas experienced intermittent snow–covered periods with field validated accuracies between -12.2 and +33.9 days.

Here is the new text in its entirety (5.1 Workflow validation):

Comparisons of the M*D10A1 derived SDDUR and SDDUR-ASWS were made over

the study period (Fig. 2). The spatially corresponding M*D10A1 pixel was compared for each ASWS station and for each hydrologic year and we calibrated our workflow using 75% of the ASWS data. The parameter values that minimized the combined MAE were a LOWESS bandwidth of 0.2, and a NDSI threshold of 30 for SDON and SDOFF. The error distributions for SDON, SDOFF and SDDUR are summarized in Fig. 2. The bias corrected MAE of 8.7, 8.9 and 13.1 days was observed for SDON, SDOFF and SDDUR, respectively. We acknowledge that other bandwidth and NDSI threshold combinations may improve results in other geographic regions or within sub-regions of BC, and is dependent on the training data used in this study. However, we caution over optimization of 10 parameters due to the inherent difference is scale between the MODIS pixels and the ASWS snow pillows. Using the remaining 25% of the ASWS stations, which we use as a validation dataset, we find bias corrected MAE of 12.7, 12.6 and 16.6 days for SDON, SDOFF and SDDUR, respectively (Fig. 2).

Here is the new text in its entirety (7.1 Snow season extraction):

Calibrating our workflow to a subset of the ASWS resulted in MAE values slightly lower than those reported in other studies (e.g., Lindsay et al., 2015).

1.4

We also add on page 14, lines 15-18 that the influence of elevation on the response of snow duration to the teleconnections in this region is generally consistent with McClung, 2013, who found that the influence of El Nino on Type 2 avalanche frequency and moisture content was less important at higher elevations.

Here is the new text in its entirety (5.5 Elevation dependency; page 14, lines 15-18):

The influence of elevation on the response of snow cover to the ocean-climate teleconnections is consistent with McClung (2013) who found that the influence of El Niño on avalanche frequency and moisture content was less important at higher elevations.

============================

RC1 Comment 2:

"Section 4.1. (MODIS processing). Why not to use also MOD10A1 data from 2001 and 2002? Will it have some impact on the results? MODIS mapping accuracy. It will be interesting to see the overall accuracy of the cloud reducing (interpolation) method (based on comparison of MODIS and ASWS). This will allow to put in more context the results of error assessment of the start, end and duration of snow cover."

Response to RC1 Comment 2:

2.1

This is a very simple, yet very important question. We have highlighted that there is some variability in the start date of the time series of studies that use MODIS time series, and have added a line (Section 3.2 page 5, lines 6-8) that highlights previous studies that use 2000, 2001, 2002, and 2003 as the beginning of their timeseries. We use the 2002 hydrological year because it is the beginning of the period of overlap between MODIS Aqua and Terra, thus we have 2 views of British Columbia per day (Section 3.2 page 5, lines 5-6). This has many advantages of having more data on our time series, and also we have only one error assessment to do. Incorporating the 1999-2001 period would require a separate error assessment as we have about half the satellite images.

Here is the new text in its entirety (3.2 MODIS snow cover product; page 5, lines 5-8):

We only use imagery acquired after 1 Sep 2002 as this is the first hydrological year where both platforms are fully operational. Other studies begin their analysis of the MODIS snow cover product in 2000 (Saavedra et al., 2018), 2001 (Hammond et al., 2018), 2002 (Dariane et al., 2017) and 2003 (Pan et al., 2018).

2.1

As for the comparison of MODIS and ASWS data, we have Figure 2 that highlights the errors between the MODIS methods and the ASWS data for our calibration (75% of

the ASWS stations) and the validation data (25% of the ASWS stations). This figure is discussed in section 5.1, where we assess our errors.

See response 3.2 above

============================

References

Allchin, M. I. and Déry, S. J.: A spatio-temporal analysis of trends in Northern Hemisphere snow-dominated area and duration, 1971–2014, Ann. Glaciol., 58, 21–35, https://doi.org/10.1017/aog.2017.47, 2017.

Barton, M.: Twenty-Seven years of manual fresh snowfall density measurements on Whistler Mountain, British Columbia, Atmosphere-Ocean, 55, 144–154, 2017.

Dariane, A. B., Khoramian, A., and Santi, E.: Investigating spatiotemporal snow cover variability via cloud-free MODIS snow cover product in Central Alborz Region, Remote Sens. Environ., 202, 152–165, 2017.

Hammond, J. C., Saavedra, F. A., and Kampf, S. K.: Global snow zone maps and trends in snow persistence 2001-2016, Int. J. Climatol., 38, 4369–4383, https://doi.org/10.1002/joc.5674, 2018.

Khoramian, A. and Dariane, A. B.: Developing a Cloud-Reduced MODIS surface reflectance product for snow cover mapping in mountainous regions, Geosci. J., 7, 29, 2017.

Monitoring snow cover variability (2000–2014) in the Hengduan Mountains based on cloud-removed MODIS products with an adaptive spatio-temporal weighted method, J. Hydrol., 551, 314–327, 2017.

Lindsay, C., Zhu, J., Miller, A. E., Kirchner, P., and Wilson, T. L.: Deriving snow cover metrics for Alaska from MODIS, Remote Sensing, 7, 12 961–12 985, https://doi.org/10.3390/rs71012961, 2015.

McClung, D. M.: The effects of El Niño and La Niña on snow and avalanche patterns in British Columbia, Canada, and central Chile, J. Glaciol., 59, 783–792, 2013.

Moreno, Á., García-Haro, F. J., Martínez, B., and Gilabert, M. A.: Noise reduction and gap filling of fAPAR time series using an adapted local regression filter, Remote Sensing, 6, 8238–8260, 2014.

Pan, C. G., Kirchner, P. B., Kimball, J. S., Kim, Y., and Du, J.: ABoVE: Rain-on-Snow frequency and distribution during cold seasons, Alaska, 2003-2016, 2018.

Saavedra, F. A., Kampf, S. K., Fassnacht, S. R., and Sibold, J. S.: Changes in Andes snow cover from MODIS data, 2000–2016, The Cryosphere, 12, 1027–1046, https://doi.org/10.5194/tc-12-1027-2018, 2018.

[Figure]

**Fig. 1.** An example of the LOWESS time series interpolation of MODIS NDSI data at the ASWS station 1A02P McBride (Upper) for the hydrological years 2002-2017.

[Figure]

**Fig. 2.** Distribution of errors calculated from the difference between MODIS and ASWS values for each snow measurement.

---

## Author Comment (AC2) · 2 Aug 2019

Response to Anonymous Referee #2 ()

Dear Anonymous Referee #2,

Thank you for your kind and thorough review of our manuscript. Your comments were well received and we believe that we have addressed all of the issues that you have pointed out. Below, we highlight, point-by-point, how we have addressed your comments with reference to the section, page number and line numbers of the modifications.

Kind regards,

On behalf of the co-authors,

Alexandre Bevington

====================

RC2 Comment 1:

"The use of LOWESS interpolation is a novel approach but is not clear what advantages has in contrast from another cloud filled approach (e.g. Li et al, 2017; Khoramian Dariane, 2017)" Response to RC2 Comment 1:

1.1

Indeed, the novel aspects of this study are somewhat hidden due to a heavier focus on the methods and results. We have added a short section titled "7.3 Significance of results" (page 17 line 32) to summarize the novel contributions of this paper. In this section we aim to highlight that we are not the first to use a LOWESS interpolation on MODIS time series data, however, this is the first study to apply it to the MODIS snow cover product. In addition the LOWESS allows us to detect the start and end dates of snow, which is an important contribution to our understanding of snow cover in British Columbia. We suggest that to extend this work further into the past, one could use lower resolution remote sensing data, or in-situ observations, and we include the provided references of (McClung, 2013, Barton 2017). These methods could likely be implemented in other regions, however the dates of the hydrological year will likely be different, and so could the optimal NDSI threshold. Finally, we highlight that the 500 m rasters that we have produced for this study could be of interest to other fields of science, and that our results could likely improve seasonal forecasting of snow.

Here is the new section in its entirety: (7.3 Significance of results):

We are not the first to use LOWESS time series interpolation on MODIS data (Moreno et al., 2014), however, to our knowledge this is the first study to use LOWESS on the MODIS snow cover product (Fig. 1). This allowed us to not only detect SDDUR from the time series, but also SDON and SDOFF. To extend this work further back in time, one could use lower spatial resolution AVHRR data (Allchin and Déry, 2017), or investigate in situ measurements (McClung, 2013; Barton, 2017). Our methods could be used anywhere that the MODIS snow cover product is present, however the definition of the hydrological year, the LOWESS bandwidth, and the NDSI thresholds may need to be optimized. Also, our method detects the longest period of continuous snow cover and will not be as useful for areas of sporadic snow cover.

The 500 m resolution annual SDON, SDOFF and SDDUR rasters produced as intermediate data in this study fill an important gap in our understanding of the regional influence of ocean-climate teleconnections on snow cover in British Columbia. These rasters may be useful for a number of other climatological and environmental processes in the fields of hydrology, ecology, and more. Operationally, our findings can be used to constrain seasonal forecasts of snow in British Columbia. For elevation 10 bins of 500 m by hydrozone, we have LLS and rS values for the relationships between snow cover and seasonal ocean-climate teleconnections.

1.2

In addition, to better explain the LOWESS, and to highlight it's functionality with the MODIS snow cover data, we have expanded on section 4.2 (page 8 lines 8-13) and added a figure of the LOWESS interpolation of NDSI timeseries data (Fig. 1). In this section we have also briefly compared our approach to other studies by (e.g. Li et al, 2017; Khoramian Dariane, 2017).

Here is the new text in its entirety (4.2 Snow season extraction; page 8 lines 8-13):

LOWESS time series interpolation has been shown to be more resistent to gaps and outliers than other similar methods in a study of the MODIS fAPAR (fraction of absorbed photosynthetic active radiation) product (Moreno et al., 2014). Other studies have proposed methods of removing clouds from MODIS snow cover data. Spatio-temporal filtering that uses a temporal probability of snow and a DEM (Li et al., 2017), and a spatial k-means interpolation with dynamic time warping (Khoramian and Dariane, 2017) are some of the innovative methods being developed in the rapidly evolving field. Our focus was to find a method that easily detects SDON and SDOFF, and the LOWESS does that quickly, efficiently and accurately.

=======================

RC2 Comment 2:

"The result shows latitudinal changes of magnitude of SDON, SDOFF, and SDDUR. Hammond et al (2018) showed a change of snow persistence in BC associated to elevation change with similar latitude and probably SD could have elevation relationship as well. I suggest running an elevation analysis to looking for elevation dependent factor in your 65 thousand locations. The elevation-latitude SD changing could be a great complement to current results."

Response to RC2 Comment 2:

2.1

Indeed, elevation (z), latitude (y) and longitude (x) are important controls on snow cover in British Columbia. The focus of our study is on the regional influence of ocean-climate teleconnections on the timing and duration of MODIS derived snow cover in British Columbia, Canada. And as such, the analysis of xyz variables as controls on snow is somewhat outside of the scope of this paper, however, we understand that completely excluding them is a lost opportunity and may cause confusion. As such, we have added a paragraph to section 5.3 (page 12, lines 9-13) and a new figure (Fig 2) where we briefly analyze how the mean snow cover changes for SDON, SDOFF, and SDDUR by XYZ.

Here is the new text in its entirety (5.3 Snow season results; page 12 lines 9-13):

In addition to summarizing the snow cover by hydrozone, we investigate the influence of latitude, longitude and elevation on the values of SDON, SDOFF and SDDUR (Fig. 2). Overall, they are all important controls on snow cover in British Columbia. Elevation is the most important terrain variable and influences SDON, SDOFF and SDDUR by -30.1, 66.9 and 97.0 days per vertical kilometer, respectively, while the influence of latitude is -3.2, 3.9 and 7.2 days per degree of latitude, respectively, and where longitude has the least important influence of 1.9, -2.4 and -4.3 days per degree of longitude, respectively (Fig. 2).

Additionally, we provide some analysis of the elevation dependency of the ONI/PDO relationship in section 5.5 (page 14, lines 7-15) and in Fig. 3.

Here is the new text in its entirety (5.5 Elevation dependency; page 14 lines 9-13):

Using rS and LLS (Fig. 3) we find that for SDON, all significant results (p < 0.05) have positive median rS and LLS (Fig. 3). Indicating that when the ONI and/or PDO are in their warm phase, SDON increases (becomes later) to a similar degree at all elevations. The range of values per elevation bin is smaller for the ONI than the PDO at low elevations. However, the number of significant relationships diminishes above 2000 m asl. Nearly all SDOFF rS and LLS values are negative for all elevations and trend towards zero at 2500 m asl. Below 2500 m asl, values decrease until 500 m asl, and increases again at 0. This suggests that lower elevations are more sensitive to changes in ONI and PDO. SDDUR has a very similar distribution by elevation as SDOFF. These results provide evidence that there are significant interactions between elevation and the ONI and PDO influence on SDOFF and SDDUR regionally over BC, with the largest magnitude and most highly correlated relationships occuring at lower elevations. The influence of elevation on the response of snow cover to the ocean-climate teleconnections is consistent with McClung (2013) who found that the influence of El Niño on avalanche frequency and moisture content was less important at higher

elevations.

========================

RC2 Comment 3:

The MODIS snow cover Collection 6 has been significantly revised and data content has been increased compared with the previous collection. For the MYD10A1 integrated a Quantitative Image Restoration (QIR) algorithm (Gladkova et al., 2012) to restore the Aqua MODIS band 6 to allow use exactly the same product for MYD10A1 and MOD10A1. I suggest including the advantages of the new collection of Aqua MODIS to highlight your novel cloud filled approach.

Response to RC2 Comment 3:

3.1

Thank you for highlighting this important research that allows for the generation of Aqua MODIS band 6 data, on which the NDSI is dependent. We have incorporated a brief explanation of the QIR in section 3.2 (page 6, lines 6-8).

Here is the new text in its entirety (3.2 MODIS snow cover product, page 6, line 6-8)

[. . .] The magnitude of this difference is exploited by the NDSI and normalized between -1 and 1, where pixels with NDSI > 0 have snow present, and those <= 0 are snow free Riggs and Hall (2015). The Aqua MODIS band 6 suffered failures shortly after launch, and as such MYD10A1 uses a reconstructed band 6 by applying a Quantitative Image Restoration (QIR) algorithm (Gladkova et al., 2012). Additionally, a series of data screens are applied to allow quality control of the NDSI results, these include data flags stored in the product as a quality assurance band (Riggs and Hall, 2015). [. . .]

========================

RC2 Comment 4:

"The NDSI threshold was 30. The previous Collection 5 had a threshold of 40 to define a pixel as snow but this fixed threshold doesn't work well in different vegetation cover condition. I suggest states the vegetation condition (as NDVI) range in your locations in order to define the limits of your NDSI threshold."

Response to RC2 Comment 4:

4.1

This is a very interesting comment. Indeed, it is difficult to use a fixed NDSI threshold with variable land cover over our study area (~1 million km2). We took an approach of having the data speak for itself. Klein et al. 1998 suggest that a single NDSI threshold (0.4) is not optimal, as it excludes snow covered regions with slightly lower NDSI values but relatively high NDVI values. Although we do not adapt the threshold based on NDVI, we optimized our workflow to ASWS stations and selected the NDSI threshold that minimized our errors; this threshold was NDSI of 30. It is important to note that this threshold is not applied to the raw data, but is dependent on the smoothing factor of the LOWESS, thus direct comparison of NDSI thresholds based on the raw data is difficult. There are many sources of potential error here, of which the most important is perhaps the bias of the ASWS locations, typically being near treeline (section 3.4, page 6 lines 21-24). Given we do not have enough data in all land cover types, we assume that this works best. We expand our description of our threshold selection in section 5.1 and have added Figure 1 (see above) to demonstrate that a threshold of 30 is reasonable.

=======================

References

Allchin, M. I. and Déry, S. J.: A spatio-temporal analysis of trends in Northern Hemisphere snow-dominated area and duration, 1971–2014, Ann. Glaciol., 58, 21–35, https://doi.org/10.1017/aog.2017.47, 2017.

Barton, M.: Twenty-Seven years of manual fresh snowfall density measurements on Whistler Mountain, British Columbia, Atmosphere-Ocean, 55, 144–154, 2017.

Dariane, A. B., Khoramian, A., and Santi, E.: Investigating spatiotemporal snow cover variability via cloud-free MODIS snow cover product in Central Alborz Region, Remote Sens. Environ., 202, 152–165, 2017.

Gladkova, I., Grossberg, M., Bonev, G., Romanov, P., and Shahriar, F.: Increasing the accuracy of MODIS/Aqua snow product using quantitative image restoration technique, IEEE Geoscience and Remote Sensing Letters, 9, 740–743, 2012.

Hammond, J. C., Saavedra, F. A., and Kampf, S. K.: Global snow zone maps and trends in snow persistence 2001-2016, Int. J. Climatol., 38, 4369–4383, https://doi.org/10.1002/joc.5674, 2018.

Khoramian, A. and Dariane, A. B.: Developing a Cloud-Reduced MODIS surface reflectance product for snow cover mapping in mountainous regions, Geosci. J., 7, 29, 2017.

Monitoring snow cover variability (2000–2014) in the Hengduan Mountains based on cloud-removed MODIS products with an adaptive spatio-temporal weighted method, J. Hydrol., 551, 314–327, 2017.

Lindsay, C., Zhu, J., Miller, A. E., Kirchner, P., and Wilson, T. L.: Deriving snow cover metrics for Alaska from MODIS, Remote Sensing, 7, 12 961–12 985, https://doi.org/10.3390/rs71012961, 2015.

McClung, D. M.: The effects of El Niño and La Niña on snow and avalanche patterns in British Columbia, Canada, and central Chile, J. Glaciol., 59, 783–792, 2013.

Moreno, Á., García-Haro, F. J., Martínez, B., and Gilabert, M. A.: Noise reduction and gap filling of fAPAR time series using an adapted local regression filter, Remote Sensing, 6, 8238–8260, 2014.

Pan, C. G., Kirchner, P. B., Kimball, J. S., Kim, Y., and Du, J.: ABoVE: Rain-on-Snow frequency and distribution during cold seasons, Alaska, 2003-2016, 2018.

Saavedra, F. A., Kampf, S. K., Fassnacht, S. R., and Sibold, J. S.: Changes in Andes snow cover from MODIS data, 2000–2016, The Cryosphere, 12, 1027–1046, https://doi.org/10.5194/tc-12-1027-2018, 2018.

[Figure]

**Fig. 1.** Example of the LOWESS temporal interpolation at the ASWS station 1A02P for the hydrological years 2002-2017.

[Figure]

**Fig. 2.** Three panel plot of the influence of elevation (A), latitude (B) and longitude (C) on the mean MODIS derived value at each random sample point.

[Figure]

**Fig. 3.** Linear least squares (LLS; top panel) and Spearman correlation coefficients (rS; bottom panel) per hydrozone for SDON, sdoff and SDDUR with ONI and PDO in 500 m elevation bins.

---

## Author Response (AR1)

Dear Dr. Marie Dumont,

We are very pleased to receive this excellent news! Thank you for your time in reviewing our manuscript and overseeing the peer review process.

Please find a track changes version of the manuscript with editorial changes highlighted in yellow with comments that associate the specific change to a reviewer comment.

Please let me know if any additional information is needed at this point.

Warm regards,

Alexandre Bevington

---

## Author Response (AR2)

Dear Dr. Marie Dumont,

We are very pleased to receive this excellent news!

Thank you for your time in reviewing our manuscript and overseeing the peer review process.

Please find the final version of the manuscript with an updated data availability link.

Please let me know if any additional information is needed at this point.

Warm regards,

Alexandre Bevington